# An oxidative metabolic pathway of 4-deoxy-L-*erythro*-5-hexoseulose uronic acid (DEHU) from alginate in an alginate-assimilating bacterium

Ryuji Nishiyama[1], Takao Ojima[1], Yuki Ohnishi[2], Yasuhiro Kumaki[3], Tomoyasu Aizawa[2] & Akira Inoue [1✉]

Alginate-assimilating bacteria degrade alginate into an unsaturated monosaccharide, which is converted into 4-deoxy-L-*erythro*-5-hexoseulose uronic acid (DEHU). DEHU is reduced to 2-keto-3-deoxy-D-gluconate by a DEHU-specific reductase using NAD(P)H. This is followed by pyruvate production via the Entner-Doudoroff pathway. Previously, we identified FlRed as a DEHU reductase in an alginate-assimilating bacterium, *Flavobacterium* sp. strain UMI-01. Here, we showed that FlRed can also catalyze the oxidation of DEHU with NAD$^+$, producing 2-keto-3-deoxy-D-glucarate (KDGR). FlRed showed a predilection for NADH and NAD$^+$ over NADPH and NADP$^+$, respectively, and the $K_m$ value for NADH was approximately 2.6-fold less than that for NAD$^+$. Furthermore, we identified two key enzymes, FlDet and FlDeg, for KDGR catabolism. FlDet was identified as an enzyme of the ribonuclease activity regulator A family, which converts KDGR to α-ketoglutaric semialdehyde (α-KGSA). FlDeg, a type II α-KGSA dehydrogenase, generated α-ketoglutaric acid by oxidizing the aldehyde group of α-KGSA using NAD(P)$^+$. Consequently, unlike the conventional DEHU reduction pathway, DEHU can be directly converted to α-ketoglutaric acid without consuming NAD(P)H. Alginate upregulated the expression of not only FlRed and two enzymes of the DEHU-reduction pathway, but also FlDet and FlDeg. These results revealed dual pathways of DEHU metabolism involving reduction or oxidation by FlRed.

[1] Graduate School of Fisheries Sciences, Hokkaido University, Hakodate, Hokkaido, Japan. [2] Faculty of Advanced Life Science, Hokkaido University, Sapporo, Hokkaido, Japan. [3] Faculty of Science, Hokkaido University, Sapporo, Hokkaido, Japan. ✉email: inouea21@fish.hokudai.ac.jp

Alginate is a heteropolysaccharide that is biosynthesized by brown algae and certain bacteria[1,2]. It consists of β-D-mannuronic acid (M) and its C5-epimer α-L-guluronic acid (G)[3–5]. These uronic acids are linearly arranged via 1,4-glycosidic bonds and form M-block or G-block, in which M or G is continuously arranged, and the MG-block, in which M and G are randomly arranged[6–8]. Brown algae are the major alginate producers in the ocean. Alginate is utilized as a component of cell wall polysaccharides[9] and is located in the intercellular matrix[10]. Some terrestrial bacteria of the genera Pseudomonas[11,12] and Azotobacter[13] also produce alginate as a component of biofilms[14–16]; however, unlike algal alginates, bacterial alginates are partially acetylated at $O_2$ and/or $O_3$ of M residues[17,18].

In nature, alginate-assimilating organisms, which incorporate alginate as the carbon source, have been isolated from both terrestrial and marine environments, for example, from bacteria and marine herbivorous mollusks such as abalone, snail, and sea hare[19–21]. They use multiple alginate lyases with different substrate specificity and mode of cleavage for complete degradation of alginate into 4-deoxy-L-erythro-5-hexoseulose uronic acid (DEHU)[22,23]. DEHU is then reduced to 2-keto-3-deoxy-D-gluconate (KDG) with NAD(P)H by a DEHU-specific reductase, which has been functionally characterized in Sphingomonas sp. strain A1[24,25], Zobellia galactanivorans[26], Flavobacterium sp. strain UMI-01[27], Vibrio splendidus[28], and abalone[29] till date. Recently, it was reported in alginate-producing organisms that the brown alga Saccharina japonica has an alginate lyase that produces DEHU[30], as well as functional DEHU reductase[31]. In alginate-assimilating bacteria, KDG is possibly utilized via the Entner-Doudoroff pathway and metabolized into pyruvate[22,32]. The process of converting alginate to pyruvate and glyceraldehyde 3-phosphate (GAP) in a single organism was biochemically reproduced using purified recombinant enzymes (three alginate lyases, the DEHU reductase FlRed, the KDG phosphatase FlKin, and the 2-keto-3-deoxy-6-phosphogluconate (KDPG) aldolase FlAld from Flavobacterium sp. strain UMI-01) in vitro[33]. One NAD(P)H molecule is unilaterally consumed to convert one DEHU to one molecule each of pyruvate and GAP during alginate metabolism. In addition, one molecule of NADH can be regenerated when one GAP molecule is oxidized to one 1,3-bisphosphoglycerate. This indicates that excessive reducing power is not generated during conversion of alginate to pyruvate when alginate is used as the sole carbon source in alginate-assimilating bacteria.

In this study, we showed that FlRed can both oxidize and reduce DEHU. Furthermore, we successfully identified two enzymes for the metabolism of the oxidation product of DEHU. These results will unveil a DEHU oxidation pathway in alginate-assimilating bacteria.

## Results

**Identification of a DEHU-oxidizing enzyme**. When the cell extract from Flavobacterium sp. strain UMI-01 was incubated with DEHU and NADH, the spot corresponding to KDG appeared after 2 min in thin layer chromatographic (TLC) analysis (Fig. 1a). FlRed catalyzes this conversion as described in our previous study[27]. The KDG spot became darker with incubation, but another spot (RP1 in Fig. 1a) with mobility lesser than that of KDG appeared after 15 min. Hence, we hypothesized that $NAD^+$ generated upon reduction of DEHU by FlRed in the cell extract acted as a coenzyme for DEHU oxidation by another enzyme. To confirm this, DEHU and the cell extract were incubated with $NAD^+$ and the reaction products were investigated. As shown in Fig. 1b, the spot showing the same mobility as RP1 was preferentially detected during incubation, and the KDG spot was less

intense than that shown in Fig. 1a. Next, we attempted to isolate the enzyme using $NAD^+$ as a coenzyme for DEHU. After four steps of column chromatography (Supplementary Fig. 1 and Supplementary Table 1), the 27 kDa protein detected using sodium dodecyl sulfate-polyacrylamide gel electrophoresis (SDS-PAGE) was purified (Fig. 1c). Its eluted position on the gel filtration chromatograph corresponded to approximately 102 kDa (Supplementary Fig. 1d), suggesting that this protein forms a tetramer in native condition. TLC analysis of products in a mixture containing this protein, DEHU, and $NAD^+$ showed generation of both RP1 and KDG after reaction (Fig. 1d). Interestingly, analyses of its N-terminal and partial amino acid sequences indicated that this protein was equivalent to the DEHU-specific reductase FlRed[27] (Fig. 1e), and other genes encoding proteins showing significant identity with these sequences were not identified in the draft genome database of the strain UMI-01.

Based on this result, recombinant FlRed (recFlRed) was generated without any additional sequence (Supplementary Fig. 2) and its activity was assayed. As expected, when the mixture of DEHU, recFlRed, and $NAD^+$ was incubated, both spots for RP1 and KDG were detected, similar to that observed in the case of the cell extract (Fig. 1f), indicating that FlRed is the dual functional enzyme with DEHU-reduction and -oxidation activities depending on the NADH and $NAD^+$ concentrations, respectively. Next, each reaction product was purified (Supplementary Fig. 3) and was subjected to an electrospray ionization-mass spectrometry (ESI-MS) analysis. KDG was the reduced product of DEHU generated by recFlRed (Fig. 1g, h) as reported previously[27]. The unknown product RP1 showed two ions of m/z 191.02 and 213.00, which were the major products in negative-mode ESI-MS spectra, and the latter ion was believed to be sodiated (Fig. 1i). Considering that RP1 is produced via the oxidation of DEHU, it was predicted to be a compound, 2-keto-3-deoxy-D-glucarate (KDGR), with a structure in which the aldehyde group of DEHU is oxidized. The structural validity was examined by carboxy group detection and ESI-MS/MS measurement.

Detection of the carboxy group was performed by the method of Endo[34], in which the absorbance at 598 nm derived from the bromophenol blue generated in the reaction solution is measured. The presence of D-glucuronic acid and α-ketoglutarate (α-KG) lowered the absorbance when compared to solutions without them (Supplementary Fig. 4). However, the presence of pyruvate, DEHU, or KDG did not change the absorbance. This indicated that although this method is used to detect carboxyl groups, it cannot detect the carboxyl group of the α-keto acid structure. RP1 showed a significantly lower absorbance at 598 nm, suggesting that it contains a carboxyl group that is not derived from the α-keto acid structure. When glucuronic acid was used as the standard, the number of carboxyl groups in α-keto acid and RP1 were calculated to be 1.1 and 0.45 per molecule, respectively. The estimated number of carboxyl groups in RP1 was less than anticipated, perhaps because some carboxyl groups formed unexpected chemical bonds, such as esters, under the reaction conditions.

The ESI-MS and MS/MS spectra of DEHU and RP1 are shown in Supplementary Fig. 5. In the MS/MS spectrum, a major fragment peak of DEHU at m/z 175.02 appeared at m/z 157.01 (Supplementary Fig. 5b). This was thought to be caused by the elimination of one water molecule from DEHU, and two probable chemical formulas, 2,5,6-trioxohexanoate and 2,4,6-trioxohexanoate, were considered as shown in Supplementary Fig. 5e. If RP1 is KDGR as predicted and was dehydrated in the same manner with DEHU during ionization, two probable structures are 5-carboxy-2,5-dioxopentanoate and 5-carboxy-2,4-dioxopentanoate that have the same molecular mass (m/z 173.01) (Supplementary Fig. 5e). However, since the latter has a β-keto

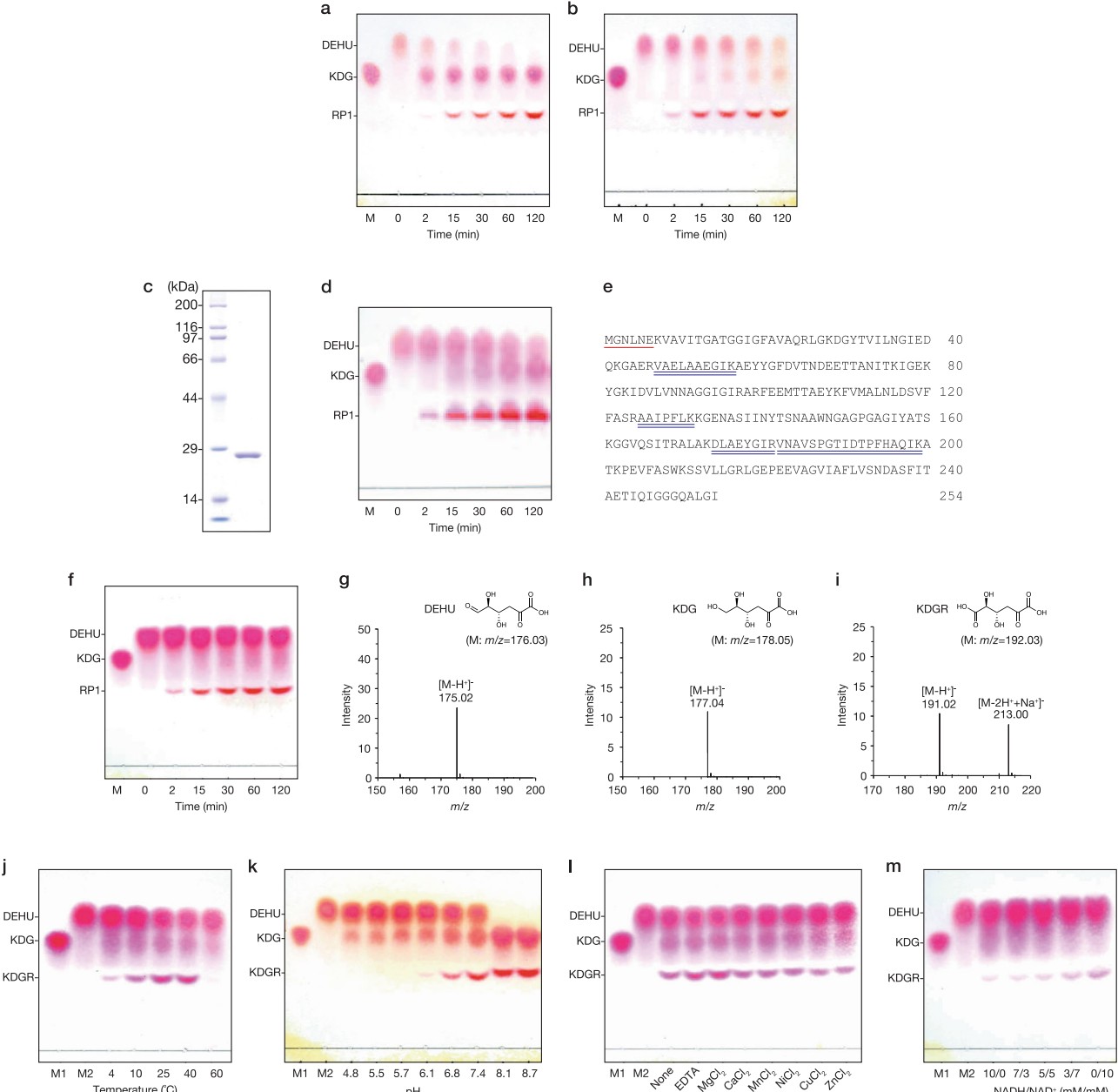

**Fig. 1 Reduction and oxidation of DEHU. a**, **b** TLC analysis of products in a mixture. Reaction was conducted in 10 mM sodium phosphate (pH 7.4), 100 mM KCl, 1 mM DTT, 0.5 mg mL$^{-1}$ cell extract, and 25 mM DEHU with 5 mM NADH (**a**) or NAD$^+$ (**b**) at 25 °C. M, KDG; *RP1*, unknown reaction product 1. **c** SDS-PAGE of purified DEHU-oxidizing enzyme from cell extract. Left lane, protein marker; right lane, purified protein after gel filtration chromatography. **d** TLC analysis of a mixture containing 50 µg mL$^{-1}$ purified protein (right lane of (**c**)), 25 mM DEHU, and 5 mM NAD$^+$. Reaction condition was same as in (**b**). M, KDG. **e** The amino acid sequence of FlRed[27]. Single and double lines represent the protein (right lane of (**c**)) sequences determined using an automated Edman sequencer and MALDI-TOF MS, respectively. **f** TLC analysis of a mixture containing 10 µg mL$^{-1}$ recFlDet, 25 mM DEHU, and 5 mM NAD$^+$. Other reaction conditions are identical to that in (**d**). M, KDG. **g–i** The negative-ion ESI mass spectrum of DEHU (**g**), KDG (**h**), and purified RP1 (**i**). Structures of DEHU, KDG, and possible structure of RP1 predicted from determined mass are shown as an inset, respectively. **j–m** The effects of temperature (**j**), pH (**k**), divalent cations (**l**), and the NADH/NAD$^+$ ratios (**m**) on recFlRed activities. Each sample was analyzed using TLC. M1 and M2 represent KDG and DEHU, respectively. In (**j**), enzyme assay was conducted in 10 mM sodium phosphate (pH 7.4), 100 mM KCl, 1 mM DTT, 20 µg mL$^{-1}$ recFlRed, and 25 mM DEHU with 5 mM NAD$^+$ for 30 min at indicated temperatures. In (**k**), the assay condition was same as in (**j**), except that the reaction was conducted in 10 mM potassium acetate (pH 4.8–5.5), 10 mM potassium phosphate (pH 5.7–7.4), and 10 mM Tris-HCl (pH 8.1–8.7) at 25 °C. In (**l**), the assay condition was same as in (**j**) except that the reaction was conducted in 1 mM of the indicated metal ions or 5 mM EDTA. "None" indicates that no divalent metal ion or EDTA was added. In (**m**), the assay condition was same as in (**j**), except that the reaction was conducted using the indicated concentrations of NADH and NAD$^+$ at 25 °C.

acid structure, it is presumed that 2,4-dioxopentanoate (*m/z* 129.02) is rapidly generated by decarboxylation (Supplementary Fig. 5e). As shown in Supplementary Fig. 5d, the major fragment in the MS/MS spectrum of RP1 was *m/z* 129.02, which is consistent with the mass of 2,4-dioxopentanoate, and no peak with a mass of 173.01 was observed. This result does not conflict with the prediction that RP1 is KDGR. Thus, we concluded that DEHU was converted to KDGR by FlRed oxidase activity.

Effects of temperature, pH, and divalent metal ions on recFlRed activities in the presence of 5 mM $NAD^+$ were investigated. DEHU was oxidized and reduced between 4 and 40 °C, but recFlRed activity was poor at 60 °C and KDGR production was not detected (Fig. 1j). Although DEHU reduction was detected at pH ranging from 4.8 to 8.7, DEHU was not oxidized under pH 5.7. The KDGR spot appeared between pH 6.1 and 8.7 (Fig. 1k). DEHU was entirely converted to KDGR and KDG at pH 8.1 and 8.7. Thus, at physiological pH (around 7.4), this enzyme was found to be capable of both reduction and oxidization of DEHU. Divalent metal ions or EDTA did not intrinsically affect recFlRed activities (Fig. 1l). The ratio of the concentrations of NADH and $NAD^+$ was an important factor that regulated FlRed function; higher NADH: $NAD^+$ ratio promoted reductase or oxidase function of FlRed (Fig. 1m).

The kinetic parameters for the reductase and oxidase activities of FlRed on DEHU were determined at 20 °C (Table 1). Although FlRed used both NADH and NADPH as coenzymes for the reduction of DEHU, the $K_m$ value for NADH was 0.037 mM, which was approximately 26.5-fold lower than that of NADPH. Additionally, it was found that FlRed could use not only $NAD^+$, but also $NADP^+$ as a coenzyme for the oxidation of DEHU with $K_m$ values of 0.098 and 1.4 mM, respectively. These results indicated that FlRed has higher affinity for the reduced form than the oxidized form of each coenzyme, and that NADH was the most preferred coenzyme among them. The $k_{cat}$ values for the reductase and oxidase activities were 160.0 $s^{-1}$ with NADH and 34.8 $s^{-1}$ with $NAD^+$, respectively, and the catalytic efficiency $k_{cat}/K_m$ of the reductase was about 12.2-fold higher than that of oxidase. On the other hand, there was no significant difference in the affinity for DEHU regardless of the form of any of the coenzymes since the $K_m$ values for DEHU were 2.0–2.7 mM. These results suggested that the ratio and concentrations of coenzymes, but not DEHU, are key factors that regulate FlRed activities.

Although we also investigated the catalytic activity of the previously identified KDG kinase FlKin[33] or the KDPG aldolase FlAld[33] for KDGR, TLC and ESI-MS analyses showed that these enzymes could not catalyze conversion of KDGR (Supplementary Fig. 6). Therefore, we attempted to isolate an enzyme catalyzing KDGR conversion from the cell extract.

**Identification of the KDGR dehydratase/decarboxylase.** We observed that the incubation of KDGR with the cell extract of strain UMI-01 resulted in a decrease in absorbance at 548 nm in the thiobarbituric acid (TBA) method (Fig. 2a). As this was possibly because of the presence of enzyme(s) responsible for the conversion of KDGR in the cell extract, we purified the native enzyme using the decrease in absorbance at 548 nm of the KDGR mixture after the TBA reaction as an index of activity. As a result, the protein, termed FlDet, estimated at approximately 34 kDa on an SDS-polyacrylamide gel was successfully purified via three-

step chromatography (Fig. 2b, Supplementary Fig. 7, and Supplementary Table 2). As the molecular weight estimated from the elution position of gel filtration chromatography was about 156 kDa, FlDet was presumed to form a tetra- or pentamer. The N-terminal 19 residues of FlDet were determined to be Met-Thr-Thr-Leu-Ser-Ala-Ser-Thr-Lys-Glu-Lys-Leu-Lys-Thr-Val-Ser-Thr-Pro-Thr, and a protein consisting of 237 residues was found as candidate FlDet by searching the draft genome database of strain UMI-01 (Supplementary Fig. 8). The calculated molecular weight of this protein was 25,697 and it was about 8 kDa smaller than the estimated one after SDS-PAGE. Next, recombinant FlDet (recFlDet) was expressed with a His-tag at its C-terminus and purified. Although the calculated molecular weight of recFlDet was 27,394 (Supplementary Fig. 9), purified recFlDet was approximately 36 kDa on an SDS-polyacrylamide gel (Fig. 2c, inset). Therefore, we measured the molecular mass of recFlDet using matrix-assisted laser desorption/ionization time-of-flight mass spectrometry (MALDI-TOF-MS) and determined it to be 27,260 (Fig. 2c), which was mostly equivalent to the calculated mass. Thus, FlDet showed abnormal mobility on an SDS-polyacrylamide gel. The amino acid sequence of FlDet showed high identity (63–80%) with those of proteins annotated as ribonuclease activity regulator A (RraA)-like protein of other alginate-assimilating bacteria such as *Flavobacterium frigidarium*, *Z. galactanivorans*, and *Flavobacterium tegetincola* (Supplementary Fig. 10). Compared to the characterized proteins in the RraA family, FlDet showed 26% and 20% identities with an RNase E inhibitor of *Streptomyces coelicolor* A3[35] and 4-hydroxy-4-methyl-2-oxoglutarate/4-carboxy-4-hydroxy-2-oxoadipate (HMG/CHA) aldolase of *Pseudomonas putida* F1[36] having an RraA-like fold, respectively. However, reports on enzymes that catalyze KDGR conversion are lacking.

After incubation of KDGR with recFlDet, the spots of KDGR and RP2 gradually disappeared and appeared, respectively, on the TLC plate (Fig. 2d). RP2 was successfully purified using an anion exchange chromatography (Supplementary Fig. 11) from the reaction mixture and was subjected to ESI-MS analysis. An ion of $m/z$ 129.02 was generated in the negative mode (Fig. 2e). As the mass of this ion was consistent with that of α-ketoglutaric semialdehyde (α-KGSA), it was believed to be caused by removal of one water and one carbon dioxide from KDGR. This structure supported the TBA reaction-mediated decrease in absorbance at 548 nm due to the absence of a *cis*-diol structure between C–4 and C–5, and also because periodate could not form β-formylpyruvate. Thus, FlDet was considered to show dehydratase and decarboxylase activities with KDGR as the substrate.

The KDGR activity of recFlDet was biochemically evaluated using the TBA method, and its optimum temperature, pH, and KCl concentration were 60 °C (Fig. 2f), 7.9 (Fig. 2g), and 10 mM (Fig. 2h), respectively. Appreciable activities of recFlDet were observed under the physiological condition, i.e., 20−30 °C, pH 7−8, and 100−150 mM KCl. As shown in Fig. 2i, certain divalent cations, such as $Mg^{2+}$, $Co^{2+}$, and $Ni^{2+}$, activated recFlDet, although the effect of $Mn^{2+}$ was weak. $Cu^{2+}$, $Ca^{2+}$, and $Zn^{2+}$ had no intrinsic effect. EDTA also did not affect its activity. Next, we investigated whether FlDet can catalyze the potential conversion of α-keto acids. DEHU, KDG, and KDPG with *cis*-diol structure were evaluated using the TBA method and TLC analysis. 4-Hydroxy-2-oxoglutarate (HOD) lacking a *cis*-diol structure was investigated using TLC analysis. As a result, KDGR was determined to be the sole α-keto acid recognized by recFlDet (Fig. 2j, k).

**Identification of α-KGSA dehydrogenase.** We next attempted to identify the native enzyme for α-KGSA from the cell lysate. TLC analysis of the mixture of the cell extract and α-KGSA indicated the

**Table 1 Kinetic parameters of the reductase and oxidase activities of FlRed.**

| Reductase activity | $K_m$ (mM) | $k_{cat}$ ($s^{-1}$) | $k_{cat}/K_m$ ($s^{-1}\cdot mM^{-1}$) |
|---|---|---|---|
| NADH | 0.037 ± 0.002 | 160.0 ± 1.5 | 4307 |
| DEHU$_{NADH}$ | 2.6 ± 0.1 | 205 ± 0.6 | 78.1 |
| NADPH | 0.98 ± 0.1 | 82.5 ± 4.1 | 84.6 |
| DEHU$_{NADPH}$ | 2.7 ± 0.2 | 87.0 ± 1.5 | 32.1 |

| Oxidase activity | $K_m$ (mM) | $k_{cat}$ ($s^{-1}$) | $k_{cat}/K_m$ ($s^{-1}\cdot mM^{-1}$) |
|---|---|---|---|
| $NAD^+$ | 0.098 ± 0.01 | 34.8 ± 1.0 | 354 |
| DEHU$_{NAD^+}$ | 2.4 ± 0.2 | 53.4 ± 1.7 | 22.4 |
| $NADP^+$ | 1.4 ± 0.2 | 24.6 ± 2.0 | 18.1 |
| DEHU$_{NADP^+}$ | 2.0 ± 0.1 | 24.9 ± 2.2 | 12.3 |

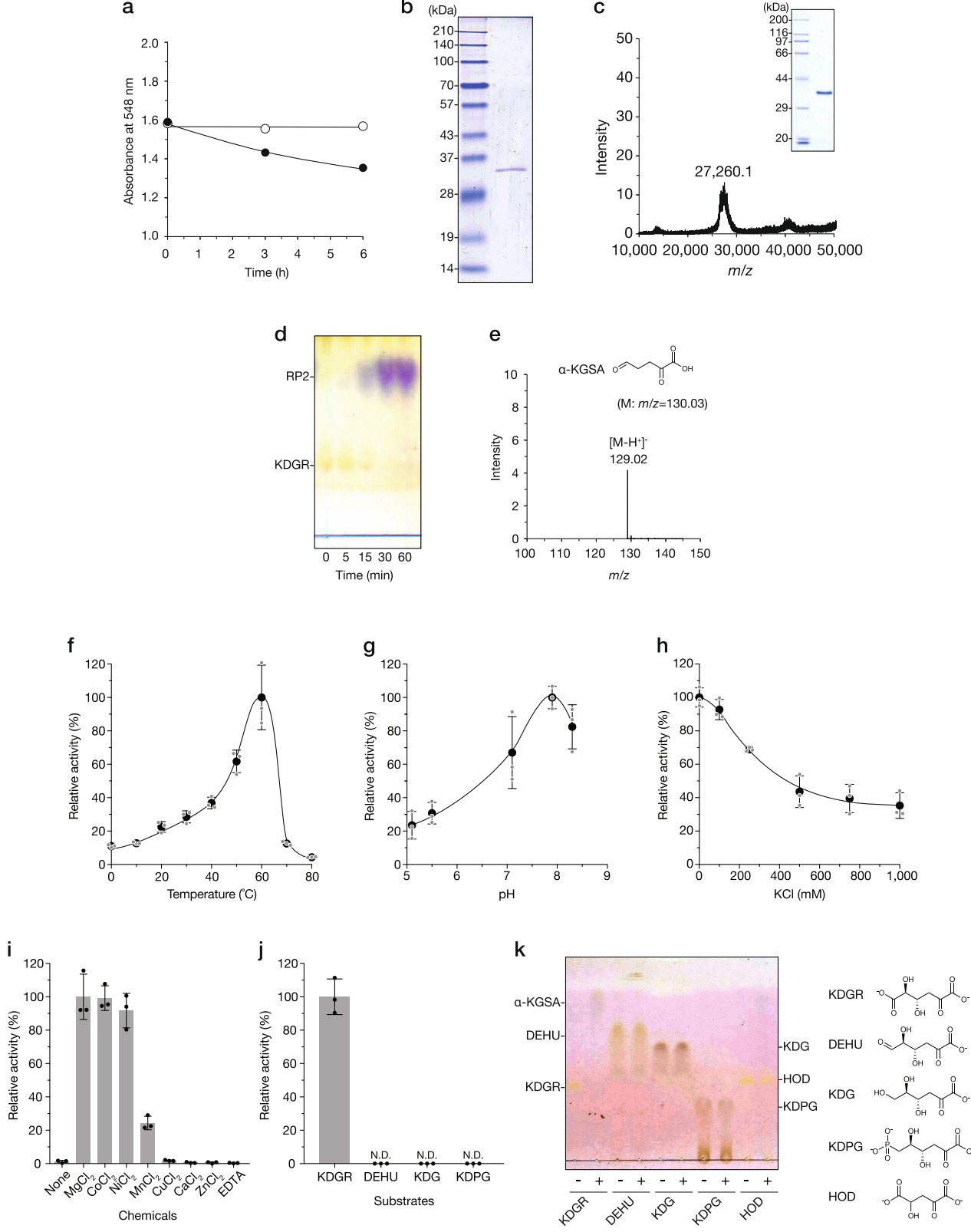

presence of at least one enzyme that uses α-KGSA as the substrate, as the α-KGSA spot disappeared after incubation (Supplementary Fig. 12). Unfortunately, we could not purify any enzyme, probably due to its low amount. Hence, we searched for a putative candidate in the draft genome sequence of strain UMI-01 (DDBJ/EMBL/GenBank accession numbers BPLU01000001–BPLU01000077). Interestingly, one gene encoding α-KGSA dehydrogenase-like protein, termed *fldeg*, was identified next to *fldet* (Fig. 3a and Supplementary Fig. 13). α-KGSA is known to be an intermediate metabolite in some organisms and is converted to α-KG by an α-KGSA dehydrogenase[37,38]. To characterize FlDeg, recombinant FlDeg (recFlDeg) was expressed with a His-tag in *Escherichia coli*. The estimated mass of the purified recFlDeg on an SDS-polyacrylamide gel was approximately 57 kDa, which is consistent with the calculated

**Fig. 2 Conversion of KDGR. a** Decrease in absorbance at 548 nm using the TBA method in the reaction mixture containing KDGR and the cell extract. Reaction was conducted in 10 mM potassium phosphate (pH 7.4), 100 mM KCl, 1 mM MgCl$_2$, 1 mM KDGR with (closed circles) or without (open circles) 0.5 mg mL$^{-1}$ cell extract at 25 °C. **b** SDS-PAGE of purified FlDet. Left lane, protein marker; right lane, purified FlDet. **c** Mass spectrum obtained using MALDI-TOF MS and SDS-PAGE (inset) of purified recFlDet. **d** TLC analysis of the reaction products of KDGR and recFlDet. Reaction was conducted in 10 mM potassium phosphate (pH 7.4), 100 mM KCl, 1 mM MgCl$_2$, 5 mM KDGR, and 3 µg mL$^{-1}$ recFlDet at 25 °C. The spots were visualized by spraying 2,4-dinitrophenyl hydrazine and heating at 40 °C for 15 min, followed by heating at 120 °C for 5 min. **e** The negative-ion ESI mass spectrum of purified RP2. The possible structure of RP2 predicted from the determined mass is shown in the inset. **f–i** The effects of temperature (**f**), pH (**g**), KCl (**h**), and divalent metal ions (**i**) on recFlDet activities. In (**f**), reaction was conducted in 10 mM potassium phosphate (pH 7.4), 100 mM KCl, 1 mM MgCl$_2$, 1 mM DTT, 1 mM KDGR, and 3 µg mL$^{-1}$ recFlDeg at the indicated temperatures for 15 min. In (**g**), the assay condition was the same as in (**f**). The reaction was conducted in 10 mM potassium acetate (pH 5.1–8.3) at 25 °C. In (**h**), the assay condition was the same as in (**f**) except that the reaction was conducted in 10–1000 mM KCl at 25 °C. In (**i**), the assay condition was the same as in (**f**), except that the reaction was conducted in the presence of 1 mM of the indicated divalent metal ions or 5 mM EDTA. "None" indicates that no divalent metal ion was added at 25 °C. Relative activity at 100% was equivalent to 9.1 U mg$^{-1}$ (**f**), 2.8 U mg$^{-1}$ (**g**), 2.2 U mg$^{-1}$ (**h**), 2.2 U mg$^{-1}$ (**i**), and 2.0 U mg$^{-1}$ (**j**), respectively. All assays were repeated thrice, and the data are shown as mean ± SD. **j, k** Substrate specificity of recFlDet. In (**i**), enzyme activities were assayed using the TBA method. Reaction was conducted in 10 mM potassium phosphate (pH 7.4), 100 mM KCl, 1 mM MgCl$_2$, 10 µg mL$^{-1}$ recFlDet, and 1 mM of each indicated substrate at 25 °C. N.D. indicates that no activity was detected. In (**j**), the assay condition was the same as in (**i**) and the reaction products were analyzed using TLC. "−" and "+" represent the incubation of each substrate without and with recFlRed for 12 h, respectively. The plate was developed under the same condition as in (**d**). The spots were visualized using the same method as in (**d**), except that heating at 120 °C was skipped. Left, TLC analysis of reaction mixture; right, each chemical structure of substrate used in (**i, j**).

**Table 2 $^{13}$C and $^1$H chemical shifts and coupling constants for standard α-KG and RP3.**

| δ$_C$ (ppm) | Standard α-KG | RP3 | δ$_H$ (ppm) | Standard α-KG | RP3 |
|---|---|---|---|---|---|
| C1$^a$ | – | – | H3 | 3.02 | 3.02 |
| C2 | 207.9 | 207.9 | H4 | 2.45 | 2.45 |
| C3 | 38.6 | 38.6 | J (Hz) | Standard α-KG | RP3 |
| C4 | 33.3 | 33.3 | H3 | 7.0 | 6.9 |
| C5 | 184.0 | 184.0 | H4 | 6.9 | 7.0 |

$^a$Chemical shift of C1 is not listed because this does not show any HMBC correlation.

mass 57,142 Da (Supplementary Fig. 14). During incubation of recFlDeg and α-KGSA in the presence of NAD$^+$ at 25 °C, the absorbance at 340 nm linearly increased and its specific activity was determined as 1.8 U mg$^{-1}$ (Supplementary Table 3). After incubation, the spot RP3 appeared similar to that of the standard α-KG on the TLC plate (Fig. 3b). Purified RP3 (Supplementary Fig. 15) was subjected to three types of analysis: an assay using the commercial glutamate dehydrogenase, ESI-MS analysis, and nuclear magnetic resonance (NMR). A glutamate dehydrogenase catalyzed the consumption of RP3 similar to that of the standard α-KG, but not DEHU, KDGR, and α-KGSA (Fig. 3c). The mass of the yielded prominent ion in negative mode was m/z 145.01, which was consistent with that of standard α-KG (Fig. 3d and Supplementary Fig. 16). Furthermore, the NMR analysis also proposed that the purified compound was α-KG (Table 2 and Supplementary Fig. 17). Therefore, we concluded that FlDeg is the enzyme catalyzing the oxidation of the aldehyde group of α-KGSA.

As shown in Supplementary Table 3, the coenzyme and substrate specificities of recFlDeg were investigated. recFlDeg preferred NAD$^+$ to NADP$^+$. α-KGSA was determined to be a better substrate than DEHU, three aldehydes (glutaraldehyde, o-phthalaldehyde, and benzaldehyde), four aldoses (glucose, galactose, arabinose, and xylose), glucuronic acid, and 2-deoxy-D-glucose. In addition, recFlDet showed significant activity under the physiological condition: its optimum temperature, pH, and KCl concentration were determined to be 25 °C, 8.2, and 100 mM, respectively (Fig. 3e–g). These results suggested that FlDet converts α-KGSA to α-KG in vivo.

Proteins homologous to FlDet were identified in other alginolytic bacteria using database search; the highest identity (61%) was observed with KGSA dehydrogenase-like protein of F.

frigidarium and 49% identity was observed with that of Z. galactanivorans (Supplementary Fig. 18). Three different types (type I, II, and III) of functional KGSADH have been identified in bacteria[38]. Alignment of FlDeg with other characterized α-KGSA dehydrogenases indicated that FlDeg belongs to type II (identity was 46%), which has higher homology than type I and III KGSADHs (identities were 21% and 18%, respectively) (Supplementary Fig. 18). Glu251 and Cys288 of FlDeg were conserved among the proteins listed in Supplementary Fig. 18. These corresponded to residues that act as a nucleophile and a general base catalyzing aldehyde dehydrogenation in the aldehyde dehydrogenase (ALDH) superfamily[22], and are known to be highly conserved in all types of KGSADHs[38].

**Expression of DEHU-metabolic enzymes in vivo.** The protein expression of FlRed, FlKin, FlAld, FlDet, and FlDeg was investigated using specific antibodies and cell extracts of strain UMI-01 (Fig. 4 and Supplementary Figs. 19 and 20). FlRed, FlKin, FlAld, and FlDeg were expressed when either glucose or alginate was used as the sole carbon source, and their expression levels tended to increase in the presence of alginate (Fig. 4b–d, f). In contrast, unlike the other enzymes, FlDet was expressed only in cells cultured in the alginate-containing medium (Fig. 4e). These results revealed that alginate upregulated the expression of the five enzymes, suggesting that the DEHU-reduction and -oxidation pathways can simultaneously function in vivo.

**A dual pathway model branched at reduction and oxidation of DEHU.** Here, we proposed a dual alginate metabolic pathway model in strain UMI-01 (Fig. 5) by summarizing the above results and those of our previous studies[27,33]. Reduction or oxidation of DEHU is a key branch point of alginate metabolism, and pyruvate or α-KG is the end product of alginate metabolism produced via DEHU-reduction and -oxidation, respectively. The intracellular concentrations of NAD(P)H/NAD(P)$^+$ and their ratio are the most important factors regulating DEHU-redox. Considering the kinetic parameters for coenzymes (Table 1), the oxidoreductase activity of FlRed seems to be dominantly regulated by the concentrations and the ratio of NADH/NAD$^+$ rather than NADPH/NADPH$^+$ in vivo.

Comparison with other alginate-assimilating bacteria at the gene level suggested that a similar dual metabolic system of DEHU may be functional in some of them. As shown in Fig. 6, alginate-assimilating bacteria harboring genes encoding high-homology proteins of FlRed, FlDet, or FlDeg were detected, especially in Flavobacteria, such as F.

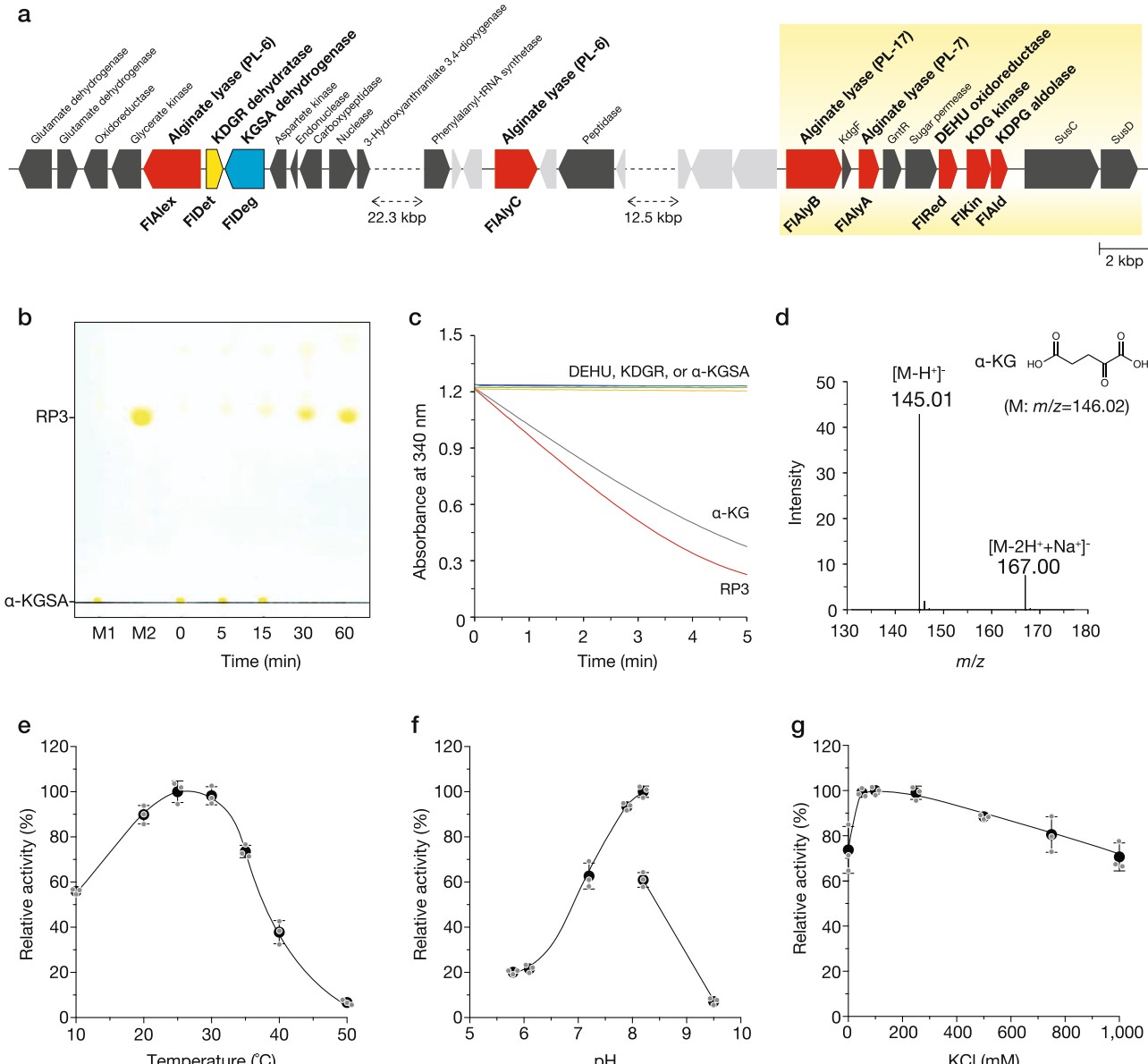

**Fig. 3 Oxidation of α-KGSA by FlDeg. a** The location of several genes encoding proteins for alginate metabolism in the genome of strain UMI-01. Genes shown as red pentagons encode characterized enzymes[27, 33, 47]. The yellow pentagon is the FlDet gene identified in this study. The blue pentagon indicated the gene for a α-KGSA dehydrogenase candidate, termed FlDeg. Dark and light gray pentagons represent genes encoding proteins, the functions of which were predicted and unpredicted from the deduced amino acid sequences, respectively. A yellow-shaded box shows the alginolytic cluster proposed in the previous studies[33, 44]. **b** TLC analysis of 2,4-dinitrophenylhydrazine derivatives. M1 and M2 represent the derivatives of α-KGSA (prepared from KDGR using FlDet and purified) and α-KG (commercially purchased), respectively. Reaction was conducted in a mixture containing 10 mM potassium phosphate (pH 7.4), 100 mM KCl, 1 mM MgCl$_2$, 1 mM DTT, 2.5 mM NAD$^+$, and 20 µg mL$^{-1}$ recFlDeg at 25 °C. The enzymatic reaction was stopped by adding one-sixth volume of 6.3 mM DNP in 3 M HCl at the indicated time and the derivatization was performed as described in "Methods". **c** Change in absorbance at 340 nm corresponding to the consumption of NADH by bovine L-glutamic dehydrogenase (GDH). Reaction was conducted in 20 mM potassium phosphate (pH 7.4), 100 mM KCl, 20 mM NH$_4$Cl, 0.2 mM NADH, and 1 U mL$^{-1}$ GDH with 0.5 mM purified RP3 (red), standard α-KG (gray), DEHU (blue), KDGR (green), or α-KGSA (yellow) at 25 °C. **d** The negative-ion ESI mass spectrum of purified RP3. A possible structure of RP3 predicted from the determined mass is shown in the inset. **e–g** The effects of temperature (**e**), pH (**f**), and KCl (**g**) on recFlDeg activities. In (**e**), reaction was conducted in 10 mM potassium phosphate (pH 7.4), 100 mM KCl, 1 mM MgCl$_2$, 1 mM DTT, 1 mM KDGR, and 3 µg mL$^{-1}$ recFlDeg at the indicated temperatures for 10 min. In (**f**), the assay condition was same as that in **e** except that the reaction was conducted with 10 mM potassium acetate (pH 5.8–8.2) or 10 mM Tris-HCl (8.2–9.5) at 25 °C. In (**g**), the assay condition was the same as that in **e** except that the reaction was conducted in various concentrations of KCl at 25 °C. Relative activity at 100% was equivalent to 1.6 U mg$^{-1}$ (**e**), 2.5 U mg$^{-1}$ (**f**), and 1.6 U mg$^{-1}$ (**g**), respectively. All assays were repeated 3 times, and the data are shown as mean ± SD.

*frigidarium, Z. galactanivorans, Cellulophaga baltica*, and *Maribacter* sp. 1_2014MBL_MicDiv. These bacteria belong to the same class as strain UMI-01 and are likely to possess a similar dual metabolic system for DEHU. In addition, this dual metabolic system was presumed to be distributed among three classes: *Polaribacter butkevichii* and *Algibacter lectus* in Flavobacteriia, *Sphingomonas koreensis* in α-proteobacteria, and *Halomonas lutea*, *Klebsiella pneumoniae*, and *Pseudoalteromonas atlantica* in γ-proteobacteria.

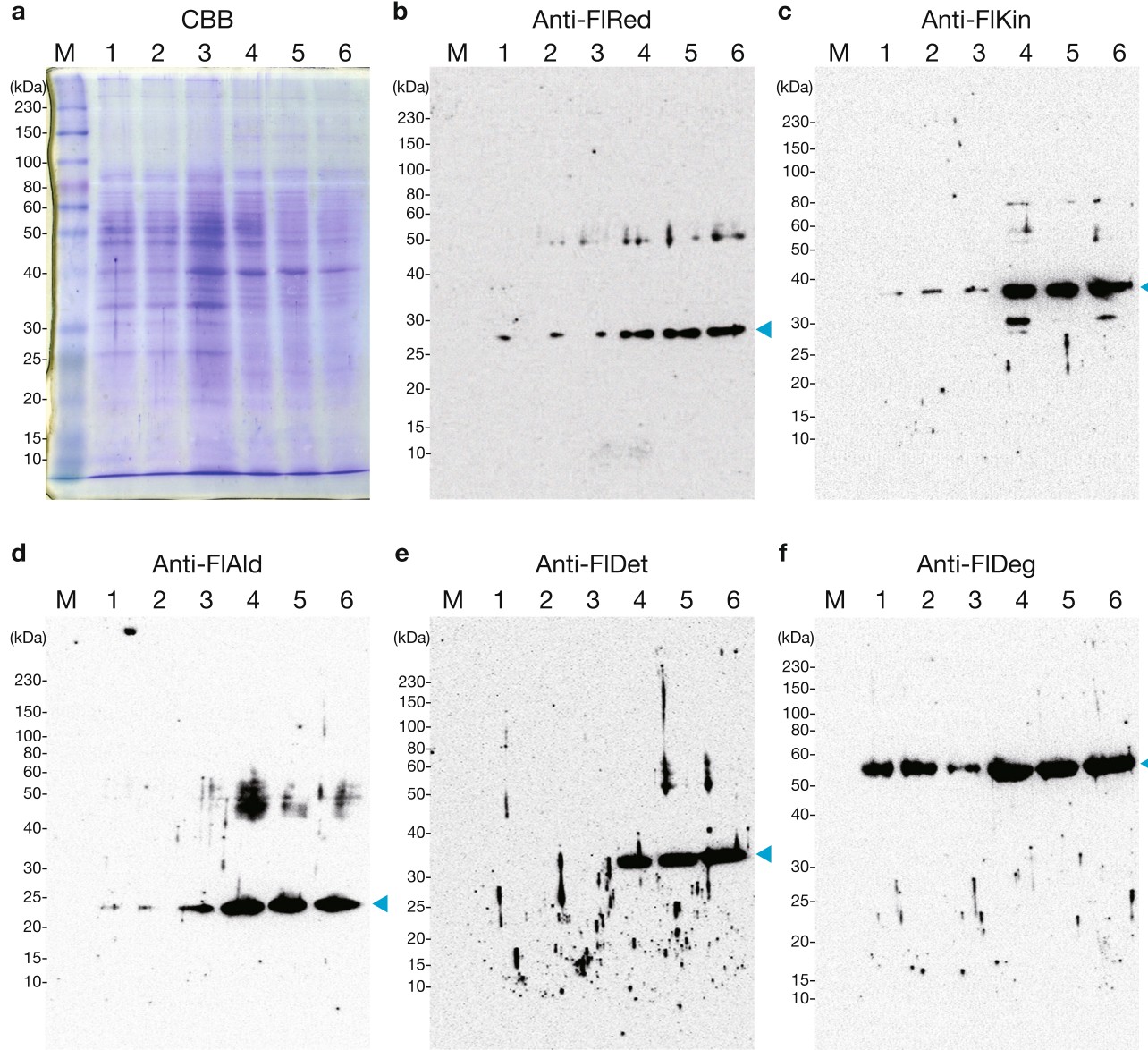

**Fig. 4 Western blot analysis of FlRed, FlKin, FlAld, FlDet, and FlDeg.** Strain UMI-01 was cultured in a medium containing glucose (lanes 1–3) or alginate (lanes 4–6) as the sole carbon source. Samples were prepared from three independent cultures for each condition. Lane M, ColorPlus Prestained Protein Ladder, Broad Range (10–230 kDa) (New England Biolabs, Ipswich, MA, USA). Each sample was applied to one gel in the order of symmetry. The gel was cut in the middle after electrophoresis, one stained with Coomassie Brilliant Blue R-250 (CBB) and the other transferred to a nitrocellulose filter for Western blotting. **a** CBB-stained gel that was electrophoresed at the same time as the gel transferred on (**b**). Uncropped and unedited CBB-stained gels for each Western blot analysis (**b**–**f**) are shown in Supplementary Fig. 19. **b**–**f** Western blot analysis using anti-FlRed, -FlKin, -FlAld, -FlDet, and -FlDeg antibodies, respectively. The cyan arrowhead indicates the expected migration position of each enzyme. The full-length gel and blots are presented. Blotted filters and uncropped images are shown in Supplementary Fig. 20.

Although several homologous enzymes showed 20–40% identity in some of the above bacteria, these bacteria might also harbor enzyme(s) that show less homology, but have the same catalytic activity. Whether other alginate-assimilating bacteria with the exception of strain UMI-01 possess a dual metabolic DEHU system will require enzymatic activity measurements at the protein level in future.

## Discussion
Here, we elucidated the dual catabolic pathway of the DEHU in strain UMI-01 and showed that its key branching step is the reduction of DEHU by FlRed. Although we have previously reported the reduction of DEHU by a C-terminal His-tagged FlRed[27], the spot corresponding to KDGR was not detected on

the TLC plate with time, unlike that shown in Fig. 1e. In this study, both native FlRed and recFlRed without any additional sequence showed reduction and oxidation activities for DEHU (Fig. 1e). We confirmed that the His-tagged FlRed formed a tetramer similar to recFlRed using gel filtration chromatography, but did not show DEHU oxidase activity in the presence of $NAD^+$ at 25 °C up to 72 h of incubation (Supplementary Fig. 21). The specific activities for DEHU reduction of native FlRed and recFlRed were determined as 164 and 173 U $mg^{-1}$ at 25 °C, which were about 30 times higher than that of the His-tagged FlRed[27]. This indicated that the fused His-tag at the C-terminus did not abolish tetramer formation, but reduced and inhibited the reductase and oxidase activities, probably due to perturbation of the binding of the substrate and/or the coenzyme.

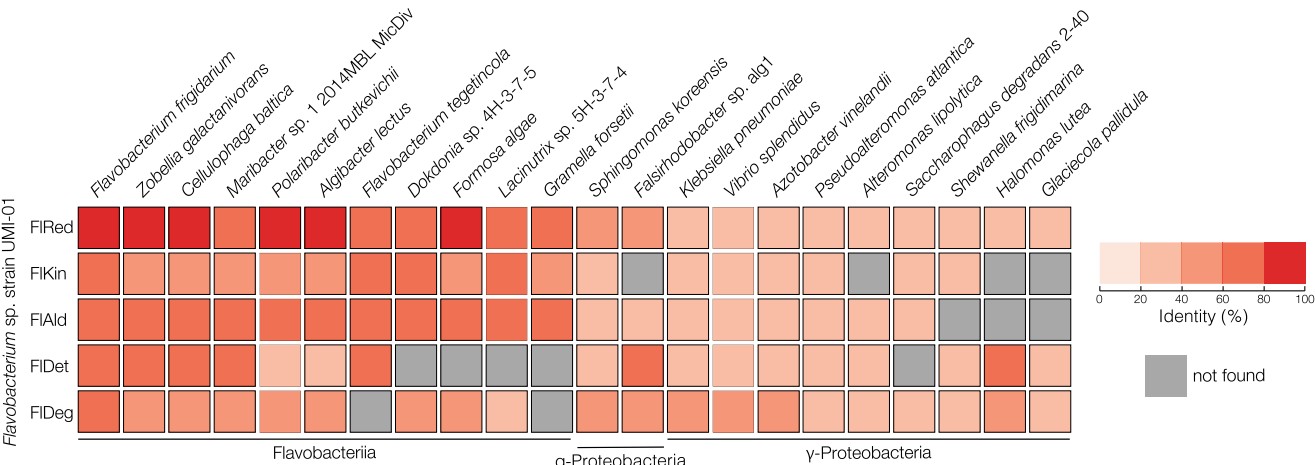

**Fig. 5 Dual metabolic pathways of DEHU in strain UMI-01.** Unsaturated monosaccharides are generated from alginate by alginate lyase(s) and are spontaneously or enzymatically[51] transformed into DEHU. DEHU is oxidized (upper) or reduced (lower) by FlRed with NAD(P)$^+$ or NAD(P)H. Final products of oxidation and reduction pathways are α-KG and pyruvate, respectively.

**Fig. 6 Enzymes involved in DEHU metabolism of strain UMI-01 and their homologous enzymes in other alginate-assimilating bacteria.** A BLASTP search was performed using the amino acid sequence of each enzyme of strain UMI-01 as the query.

Hitherto, enzyme activities of DEHU reductases from bacteria[24–28], an abalone[29], and a brown alga[31] were investigated, but there are no reports on their DEHU-oxidation activities. All recombinant DEHU reductases except A1-R and A1-R' of *Sphingomonas* sp. strain A1 were expressed as His-tagged proteins and assayed. In the present study, the addition of a His-tag to FlRed caused a loss of DEHU oxidative activity (Supplementary Fig. 21). This result proposed that the recombinant protein whose sequence is completely identical to the corresponding native protein should be used for the DEHU oxidoreductase assay. It was also reported that native A1-R[25], A1-R'[24], and abalone HdRed[29] were successfully purified and recombinant A1-R[25] and A1-R'[24] without fusion sequence were also expressed. However, the effects of NAD(P)$^+$ when using DEHU as a substrate were not investigated in either case. Therefore, to determine if DEHU reductase has DEHU oxidative activity, not only NAD(P)H, but also NAD(P)$^+$ as coenzymes should be evaluated with the use of a protein having the appropriate primary structure.

Our results demonstrated that the oxidation pathway has to be considered for the introduction of DEHU metabolism in non-alginate-assimilating microorganisms. Conventionally, as shown in Fig. 5, DEHU was believed to be converted to pyruvate only via the reduction pathway and no excess reducing power is produced in this process. Till date, some engineered microorganisms aimed at converting alginate to ethanol have been created based on this concept[39,40]. For example, a recombinant yeast was constructed by introducing foreign genes involved in

DEHU metabolism, and it produced ethanol in a medium containing both DEHU and mannitol[40]. In this study, mannitol was thought to be not only a resource for ethanol along with DEHU, but also to produce excess reduction equivalents during its metabolic processes. In this construct, mannitol-2-dehydrogenase, which oxidizes mannitol into D-fructose using NAD$^+$ as a coenzyme to produce NADH, was overexpressed along with *Vibrio harveyi* DEHU reductase, which has 36.1% identity to FlRed, in the recombinant yeast. Excess NADH was considered to be utilized in converting pyruvate derived from DEHU to ethanol. If the introduced DEHU reductase has both reductive and oxidative activities for DEHU and a higher affinity for NADH than NAD$^+$, similar to FlRed, consumption of NAD$^+$ by the oxidation of mannitol might suppress the oxidation of DEHU and promote the conversion of DEHU to KDG as much as possible. Accordingly, mannitol-2-dehydrogenase may have contributed to the generation of the excess reducing power and to the efficiency of the DEHU-reduction pathway in the recombinant yeast.

Studies have shown that some bacteria, such as *Agrobacterium tumefaciens* and *Pseudomonas syringae*, can oxidize the hexuronic acid, D-galacturonic acid, and that 2-keto-3-deoxy-L-*threo*-hexarate (5-keto-4-deoxy-D-galactarate) is generated as an intermediate[41–43]. This is hydrated and decarboxylated by 5-keto-4-deoxy-D-glucarate dehydratase/decarboxylase and converted to α-KGSA. The final product is α-KG, which results from the conversion of α-KGSA by α-KGSA dehydrogenase and NAD$^+$, as shown in this study. Although it is unclear

whether 5-keto-4-deoxy-D-glucarate dehydratase/decarboxylase recognizes 2-keto-3-deoxy-L-*erythro*-hexarate, which is the KDGR derived from DEHU, FlDet does not show significant homologies with known bacterial 5-keto-4-deoxy-D-glucarate dehydratase/decarboxylases. Till date, KDGR dehydratase/decarboxylase activity has not been detected in the RraA family, suggesting that FlDet is a unique enzyme with dehydratase and decarboxylase activities.

Two genes encoding FlDet and FlDeg were located far away from the conventional alginolytic cluster (Fig. 3a). Interestingly, *flalex*, encoding an exo-type G-specific alginate lyase that produces DEHU[44], was flanked by *fldet*. This consecutive region is about 51 kb away from the previously proposed alginolytic cluster comprising the genes for two alginate lyases (an endo-type alginate lyase FlAlyA, and an exo-type M-specific alginate lyase FlAlyB), FlRed, FlKin, FlAld, KdgF-like protein, GntR-like protein, and sugar permease-like protein[27,33]. The co-expression of FlDet and FlDeg is key for the DEHU oxidative pathway; however, the direction of transcription of *fldet* is opposite to that of *fldeg*, indicating that these genes cannot be transcribed polycistronically. At the protein level, FlDeg was expressed in small amounts, as was the case for FlRed, FlKin, and FlAld, in the glucose medium, and alginate enhanced the expression of FlDeg. On the other hand, FlDet expression could not be detected in the glucose medium, but it was detected only in the presence of alginate (Fig. 4). This result indicates that the regulation of FlDet expression may be distinct from that of FlRed, FlKin, FlAld, or FlDeg. Although the regulation mechanism of FlDet expression remains unclear, it is likely that FlDet expression is strictly regulated by DEHU and/or its metabolites, unlike other DEHU metabolism-related enzymes. To better understand the alginate-assimilation system of strain UMI-01 in detail, it will be necessary to elucidate the transcriptional regulatory mechanism of three adjacent genes, *fldet*, *fldeg*, and *flalex*, which do not belong to the conventional alginate cluster.

One of the advantages of the conversion of DEHU to α-KG circumventing pyruvate would be the efficient rotation of the citric acid cycle. The process of conversion of DEHU to α-KG without consuming NADH and activation of the citric acid cycle will increase the intracellular reducing power. α-KG is also known as a precursor of glutamate and its conversion is catalyzed by glutamate dehydrogenase. This enzyme is widely distributed in prokaryotes and eukaryotes, and two candidate genes were identified in strain UMI-01 using homology search. Glutamate is also converted to glutamine by a glutamine synthase (GS), and one candidate gene for class I type of GS was present in strain UMI-01. Considering that this strain was isolated from the rotten brown alga washed ashore, it is presumed that this bacterium naturally inhabits environments of fluctuating osmotic pressure. Till date, osmolytes in bacteria, such as amino acids (glutamine, glutamate, and proline), amino acid derivatives (betaine and ectoine), polyhydric alcohols (glycerol and mannitol), and oligosaccharides (trehalose), have been investigated[45,46]. We observed that strain UMI-01 was able to grow even in artificial seawater (without $Ca^{2+}$ to avoid alginate gelation) in which alginate was added as the sole carbon source (Supplementary Fig. 22). Thus, this bacterium is resistant to high salt concentrations, and glutamate and/or glutamine derived from α-KG may be used as osmolytes in such conditions.

## Methods

**Materials**. *E. coli* strain DH5α and BL21(DE3) were purchased from Nippon Gene (Tokyo, Japan) and were used for cloning and recombinant protein expression, respectively. A cloning vector, pTac-1, was purchased from BioDynamics (Tokyo, Japan). A modified pCold vector (TaKaRa, Shiga, Japan)[47] was used for protein expression. Reagents were purchased from FUJIFILM Wako Pure Chemical

Corporation (Osaka, Japan) unless stated otherwise. Protein concentration was determined with a TaKaRa bicinchoninic acid (BCA) protein assay kit or TaKaRa Bradford protein assay kit.

**Total cell extract from *Flavobacterium* sp. UMI-01**. *Flavobacterium* sp. strain UMI-01 was cultured in 500 mL alginate-minimum salt medium containing 1% (w/v) sodium alginate, 0.38% (w/v) $Na_2HPO_4$, 0.27% (w/v) $KH_2PO_4$, 0.036% (w/v) $NH_4Cl$, 0.02% (w/v) $MgCl_2$, and 0.1% (v/v) of trace element solution (0.97% (w/v) $FeCl_3$, 0.78% (w/v) $CaCl_2$, 0.02% (w/v) $CoCl_2·6H_2O$, $CuSO_4·5H_2O$, 0.01% (w/v) $NiCl_2·6H_2O$, and 0.01% $CrCl_3·6H_2O$ in 0.1 N HCl) at 25 °C until the $OD_{600}$ value was 1.5–1.8. Cells were harvested by centrifugation at 12,500*g* for 5 min at 4 °C and were sonicated (30 s, 6 times) with an ULTRASONIC homogenizer VP-050 (TAITEC, Saitama, Japan) at 20 kHz, 25 W in 20 mL of 10 mM sodium phosphate (pH 7.4). After centrifugation at 22,000*g* for 15 min at 4 °C, the supernatant was dialyzed against 10 mM sodium phosphate (pH 7.4).

**Purification of native FlRed**. Total cell extract of strain UMI-01 was prepared as described above except that the culture medium and the sonication buffer were 2 L and 80 mL, respectively. Ammonium sulfate was added to the total cell extract to a final concentration of 1.0 M and the supernatant was subjected to TOYOPEARL Butyl–650 M column (2.4 × 20 cm, Tosoh Bioscience, Tokyo, Japan) chromatography after centrifugation at 26,000*g* for 10 min at 4 °C. The absorbed proteins were eluted using the linear gradient of 1.0–0 M ammonium sulfate. Enzyme activity was assayed using TLC analysis as described and the active fractions were dialyzed against 10 mM sodium phosphate (pH 7.4). After centrifugation at 26,000*g* for 10 min at 4 °C, the samples were applied to a TOYOPEARL SuperQ–650 S column (2.4 × 22 cm, Tosoh Bioscience) and developed with a linear gradient of 0–0.3 M NaCl in 10 mM sodium phosphate (pH 7.4). Proteins in the active fractions were precipitated with 80% $(NH_4)_2SO_4$ and collected via centrifugation at 12,500*g* for 10 min at 4 °C. The precipitates were dissolved in 2 mL of 10 mM sodium phosphate (pH 7.4) and were dialyzed against the same buffer. The dialysate was centrifuged at 26,000*g* for 10 min at 4 °C and the supernatant was subjected to AKTA-FPLC (GE Healthcare, Chicago, IL, USA) equipped with MonoQ 4.6/100 PE column (GE Healthcare) pre-equilibrated with 10 mM sodium phosphate (pH 7.4). The developing condition was identical to that of the previous step and the active fractions were pooled and concentrated to 50 µL using a VIVASPIN20 centrifugal concentrator (Sartorius AG, Göttingen, Germany). The concentrate was applied to a Superdex 200 10/300 GL gel filtration column, in which 10 mM sodium phosphate (pH 7.4) and 100 mM NaCl were used as eluents. Molecular weight was estimated using the HMW gel filtration calibration kit (GE Healthcare).

**Purification of native FlDet**. Native FlDeg was purified using the same method as described in "Purification of native FlRed" except that the TOYOPEARL SuperQ–650 S column chromatography was omitted. FlDet activity of each fraction was monitored using the TBA method as described in "Assay for FlDet activity".

**Cloning of genes encoding FlDet and FlDeg**. Target DNA was amplified with a primer set (Supplementary Table 4) and KOD-Plus-Neo DNA polymerase (Toyobo, Osaka, Japan) using the genomic DNA of strain UMI-01 as the template. The amplified DNA was subcloned into a pTac-1 vector after treatment with an A-attachment mix (Toyobo). The recombinant plasmid was sequenced using the 3130*xl* genetic analyzer (Applied Biosystems, Foster City, CA, USA).

**Bacterial expression of recombinant enzymes**. The DNA encoding FlRed, FlDet, or FlDeg was amplified with specific primer sets (Supplementary Table 4) and KOD-Plus-Neo DNA polymerase (Toyobo). After purification of the amplified DNA with the ISOSPIN agarose gel kit (Nippon Gene), DNA was ligated into a *Nco*I/*Bam*HI site of a modified pCold vector using an In-Fusion HD cloning kit (TaKaRa). After DNA sequencing using the described method, the recombinant plasmid was introduced in *E. coli* BL21(DE3). The transformants were cultured in 1 L of 2× YT medium (1.6% Bacto tryptone, 1.0% yeast extract, and 0.5% NaCl) supplemented with 50 µg mL$^{-1}$ ampicillin at 37 °C for 16 h. After incubation at 15 °C for 2 h, 1 mL of 100 mM isopropyl β-D-1-thiogalactopyranoside (IPTG) was added. After incubation at 15 °C for 6 h, the cells were harvested via centrifugation at 12,000*g* for 15 min at 4 °C. All cells, except those producing recFlRed, were sonicated (30 s, 8 times at 20 kHz, 25 W) in a buffer containing 20 mM sodium phosphate (pH 7.4), 0.5 M NaCl, 1% Triton X-100, and 1 mg mL$^{-1}$ lysozyme. recFlDet or recFlDeg was purified using Ni-NTA agarose affinity chromatography (Qiagen, Hilden, Germany). The resin was washed with 30 mM imidazole-HCl (pH 7.4) and 0.5 M KCl, and the protein was eluted with 150 mM imidazole-HCl (pH 7.4) and 0.5 M KCl. All purified recombinant proteins were dialyzed against a buffer containing 10 mM sodium phosphate (pH 7.4) and 0.1 M KCl. For purification of recFlRed, harvested cells were sonicated with 10 mM sodium phosphate (pH 7.4) and were centrifuged at 22,000*g* for 15 min at 4 °C. Ammonium sulfate was added to the supernatant at a final concentration of 1.0 M, followed by centrifugation at 26,000*g* for 10 min at 4 °C. Then, the supernatant was subjected to chromatography using a TOYOPEARL Butyl–650 M column (2.4 × 20 cm); the absorbed proteins were eluted using the linear gradient of 1.0–0 M ammonium sulfate. DEHU reductase activities of the fractions were assayed by TLC analysis

and the active fractions were dialyzed against 10 mM sodium phosphate (pH 7.4). After centrifugation at 26,000$g$ for 10 min at 4 °C, the samples were loaded onto a TOYOPEARL SuperQ–650 S column (2.4 × 22 cm) and developed with a linear gradient of 0–0.3 M NaCl in 10 mM sodium phosphate (pH 7.4). Proteins in the active fractions were concentrated by replacing the buffer with that containing 10 mM sodium phosphate (pH 7.4) and 0.1 M KCl using the Vivaspin 20-10 K system (GE Healthcare).

**Protein sequence analysis.** The N-terminal sequence was determined using Edman degradation method with a Procise 492-HT protein sequencing system (Applied Biosystems). The internal sequences were analyzed using a Proteomics 4700 MALDI-TOF/TOF mass spectrometer (Applied Biosystems) after tryptic digestion of the sample.

**Determination of the molecular mass of recFlDet.** Purified recFlDet in distilled water was mixed with three volumes of saturated α-cyano-4-hydroxycinnamic acid in 70% acetonitrile containing 0.1% trifluoroacetic acid. The mixture was spotted on a sample plate and analyzed in positive-ion mode with a Proteomics 4700 MALDI-TOF/TOF mass spectrometer.

**Assay for FlRed activity.** DEHU was prepared from alginate using recombinant alginate lyases FlAlyA, FlAlyB, and FlAlex, as described previously[33,44]. An enzyme assay was basically performed in a reaction mixture containing 10 mM sodium phosphate (pH 7.4), 100 mM KCl, 1 mM DTT, 25 mM DEHU, and 0.5 mg mL$^{-1}$ cell extract or 10–50 μg mL$^{-1}$ purified enzyme with 5 mM NADH or NAD$^+$ at 25 °C. The enzyme reaction was terminated by adding an equivalent amount of chloroform. After vortex-mixing and centrifugation at 10,000$g$ for 5 min, the aqueous phase was subjected to TLC analysis using the TLC silica gel 60 plate (Merck KGaA, Darmstadt, Germany). Products were developed with 1-butanol: acetone: acetic acid: water (35:35:7:23, v:v:v:v) and were visualized by spraying 4.5% (w/v) TBA after the periodic acid oxidation[19].

**Determination of the kinetic parameters for FlRed.** The kinetic parameters of coenzymes and DEHU for FlRed were determined using recFlRed at 20 °C. The reaction mixture for the reductase assay to determine the parameters of NAD(P)H consisted of 10 mM sodium phosphate (pH 7.4), 0.15 M KCl, 5 mM DEHU, and 0–0.5 mM NAD(P)H. The reaction conditions for the oxidase assay to determine the parameters of NAD(P)$^+$ were the same, except 0–2.0 mM NAD(P)$^+$ was used instead of NAD(P)H. To determine the parameters of DEHU, the assay conditions were the same, except 0.2 mM NADH, 2.0 mM NADPH, 0.5 mM NAD$^+$, or 3.0 mM NADP$^+$ was used in the presence of 0–20 mM DEHU. The enzymatic reaction was stopped at 0.5, 2, 5, 10, and 15 min by adding an equal amount of chloroform. After centrifugation at 10,000$g$ for 10 min, the absorbance of the aqueous layer was measured at 340 nm. The aqueous layer was diluted with distilled water as necessary. All assays were repeated thrice. One unit (U) was defined as the amount of enzyme that consumed 1 μmol of each coenzyme in 1 min. The kinetic parameters, $K_m$ and $k_{cat}$, for each coenzyme or DEHU were determined using the Michaelis–Menten equation with Prism 8 software (GraphPad Software, San Diego, CA, USA).

**Assay for FlDet activity.** Enzyme reaction was conducted in 10 mM potassium phosphate (pH 7.4), 100 mM KCl, 1 mM MgCl$_2$, 1 mM DTT, 1 mM KDGR, and 0.5 mg mL$^{-1}$ cell extract or 3 μg mL$^{-1}$ recFlDet at 25 °C, unless mentioned otherwise. After incubation, the reaction was stopped by mixing with an equal volume of chloroform. After centrifugation at 12,000$g$ for 5 min, the aqueous layer was used for TLC analysis or TBA reaction[19,48]. TLC analysis was conducted as described above, except that the products were visualized by spraying 50 mg mL$^{-1}$ 2,4-dinitrophenyl hydrazine dissolved in a solution of sulfuric acid: ethanol: water (3:12:5, v:v:v) and heating at 40 °C for 15 min, followed by heating at 120 °C for 5 min unless otherwise noted. The decrease in absorbance at 548 nm due to TBA reaction, accompanied with the enzyme reaction, was also used to evaluate the activity of FlDet. One unit (U) was defined as the amount of enzyme that reduces 1 μmol of KDGR for 1 min. The decrease in KDGR was calculated from the decrease in β-formyl pyruvate ($\varepsilon = 29.0 \times 10^3$ M$^{-1}$ cm$^{-1}$).

**Assay for FlDeg activity.** α-KGSA (1 mM) was incubated in the reaction mixture containing 10 mM potassium phosphate (pH 7.4), 100 mM KCl, 1 mM MgCl$_2$, 1 mM DTT, 0.5 mM NAD$^+$, and 0.1 mg mL$^{-1}$ recFlDeg. Absorbance at 340 nm was monitored and one unit (U) was defined as the amount of enzyme that consumes 1 μmol of NAD$^+$ for 1 min. The reaction products were analyzed using TLC as follows. The sample was mixed with one-sixth the volume of 6.3 mM 2,4-dinitrophenyl hydrazine in 3 M HCl and incubated at 50 °C for 45 min. The derived products were extracted from the aqueous layer with an equal volume of ethyl acetate and evaporated in vacuo. The yellow solids were dissolved in ethyl acetate and subjected to TLC analysis. The samples were developed using the TLC silica gel 60 plate with 1-butanol: ethanol: 0.5 M acetic acid (7:1:2, v:v:v).

**Determination of α-keto acid.** The levels of α-keto acids, such as DEHU, KDG, KDGR, and α-KG, were determined using the spectrophotometric semicarbazide method[37,49]. The sample was mixed with the same volume of 0.1 M semicarbazide hydrochloride-0.11 M sodium acetate and incubated at 30 °C for 30 min. The α-keto acid derived from the semicarbazide derivative was quantified by measuring the absorbance at 250 nm ($\varepsilon = 7200$ (M$^{-1}$)).

**ESI-MS and -MS/MS for studying enzyme reaction products.** The purified enzyme reaction products were analyzed using ESI-MS. Each product was desalted using gel filtration with the Superdex Peptide 3.2/300 column (GE Healthcare) pre-equilibrated with distilled water and was diluted with methanol. Samples were analyzed using Exactive (Thermo Fisher Scientific, Waltham, MA, USA) in a negative-ion mode and the spectrum was scanned from 0 to 1000 $m/z$.

**Purification of the reaction product of FlRed.** KDGR was prepared in a mixture containing 10 mM potassium phosphate (pH 7.4), 25 mM DEHU, 5 mM NAD$^+$, and 50 μg mL$^{-1}$ recFlRed at 25 °C for 12 h. Then, three volumes of 2-propanol were added and the solution was incubated at −20 °C for 2 h. After centrifugation at 26,000$g$ for 10 min at 4 °C, the supernatant was evaporated in vacuo and the solid residue was dissolved in distilled water. Then, KDG and KDGR were separated using a TOYOPEARL SuperQ-650S column (2.4 × 22 cm, pre-equilibrated with distilled water) chromatography with a linear gradient of 0–0.3 M NaCl. After TLC analysis, KDG and KDGR concentrations were determined using the semicarbazole method.

**Purification of the reaction product of FlDet.** Purified KDGR (20 mM) was incubated in 10 mM potassium phosphate (pH 7.4), 1 mM MgCl$_2$, and 50 μg mL$^{-1}$ recFlDet at 30 °C. After 2 h, one-fourth volume (of the reaction mixture) of chloroform was added to stop the enzyme reaction. After centrifugation at 22,000$g$ for 10 min at 4 °C, the aqueous layer was recovered and diluted 10-fold with distilled water. Then, KGSA was purified using chromatography similar to that used for KDGR, but with a linear gradient of 0–0.2 M NaCl. KGSA concentration was also measured using a method similar to that used for KDGR. KDGR (1.5 mM) was used as the substrate for recFlDeg (20 μg mL$^{-1}$) in 10 mM potassium phosphate (pH 7.4), 100 mM KCl, 1 mM MgCl$_2$, 1 mM DTT, and 2.5 mM NAD$^+$ at 25 °C for 16 h. Four volumes of 2-propanol were added and the subsequent steps were performed in a manner similar to that used for the purification of KDGR, except that a linear gradient of 0–0.2 M NaCl was used for the chromatography.

**Purification and identification of the reaction product of FlDeg.** The reaction mixture containing 10 mM potassium phosphate (pH 7.4), 1 mM MgCl$_2$, 5 mM NAD$^+$, 1.5 mM KGSA, and 30 μg mL$^{-1}$ recFlDeg was incubated at 25 °C for 16 h, followed by the addition of 4 volumes of 2-propanol. After incubation at −20 °C for 2 h, the supernatant was obtained after centrifugation at 26,000$g$ for 10 min in vacuo and was concentrated using a rotary evaporator in vacuo. Thereafter, it was diluted 10 times with distilled water and loaded on a TOYO-PEARL SuperQ-650S column (2.4 × 22 cm) pre-equilibrated with distilled water. A linear gradient of 0–0.2 M NaCl was used for elution. Each fraction was investigated using the semicarbazide method. The purified reaction product was analyzed using three methods: (1) ESI-MS analysis was performed as described above; (2) for the enzymatic assay, reaction was performed in a mixture containing 20 mM potassium phosphate (pH 7.4), 100 mM KCl, 0.2 mM NADH, 20 mM NH$_4$Cl, 1 U mL$^{-1}$ bovine glutamate dehydrogenase (Sigma-Aldrich, St Louis, MO, USA), and 0.5 mM purified RP3, DEHU, KDGR, KGSA, or standard α-KG at 25 °C. One unit is the amount of enzyme that reduces 1.0 μmol α-KG to L-glutamic acid for 1 min; and (3) $^1$H -$^{13}$C HMBC spectra of standard α-KG or the purified product RP3 were measured at 298 K on a Bruker AVANCE III HD 600 MHz spectrometer (Bruker, Billerica, MA, USA) equipped with TXI triple resonance 5 mm probe with pulsed field gradient coil. The number of complex points were 4096 in the $t_2$ dimension and 1024 in the $t_1$ dimension. The spectral widths were 7.2 kHz in the $f_2$ dimension and 34.7 kHz in the $f_1$ dimension. The number of transients per free induction decay was 8. Delay for the evolution of $^1$H–$^{13}$C long-range scalar coupling was optimized to 8 Hz. Presaturation was performed for water suppression during relaxation delay of 5 s. $^1$H chemical shifts were adjusted with an external reference sample of trimethylsilyl propanoic acid (TSP) in water (10% D$_2$O). $^{13}$C chemical shifts, derived from HMBC spectra, were referenced indirectly to TSP[50]. Spectra were processed and analyzed using the JEOL Delta version 5 software (JEOL, Tokyo, Japan).

**Detection of the carboxy group using 4-nitrophynacyl bromide.** Detection of the carboxy group was performed by the method of Endo[34] with a slight modification. D-glucuronate, pyruvate, α-KG, DEHU, KDG, or RP1 was dissolved in 50 mM KH$_2$PO$_4$ to a final concentration of 2 mM. Each solution or 50 mM KH$_2$PO$_4$ (200 μL each) was mixed with 100 μL of an acetone solution of 10 mM 4-nitrophenacyl bromide, followed by incubation at 20 °C for 1 h. Next, 10 μL of 1.0 mg mL$^{-1}$ chloramine-T and 10 μL of 0.02 mg mL$^{-1}$ phenol red were added to each sample. The samples were then incubated at 20 °C for 30 min, and the absorbance of each was measured at 598 nm.

**Western blot analysis**. Strain UMI-01 was cultured in the minimum salt medium containing 2% (w/v) glucose or 1% (w/v) alginate as the sole carbon source at 25 °C. When the $OD_{600}$ of the culture medium was between 0.6 and 0.8, the cells were harvested by centrifugation at 12,000$g$ for 15 min at 4 °C, and soluble proteins were obtained using BugBuster Master Mix (Merck Millipore, Burlington, MA, USA) according to the manufacturer's protocol. Each sample (5 μg) was subjected to SDS-PAGE and electroblotted onto a ClearTrans nitrocellulose membrane (FUJIFILM Wako Pure Chemical Corporation). The anti-FlRed, -FlKin, -FlAld, -FlDet, and -FlDeg antibodies were raised in rabbits against the synthetic peptides CKGAERVAELAAEGI, CYGLNEEPLDNQRAL, CSKLIKPESDGNFDL, CLVMDSRKDPRAASA, and CDTAQPNRTPLPKSD, respectively, using keyhole limpet hemocyanin as a carrier protein (Eurofins Genomics, Tokyo, Japan). Each antibody was purified by affinity chromatography using an immobilized antigen column (Cellufine Formyl, JNC Corporation, Tokyo, Japan) and used as the primary antibody. A horseradish peroxidase-conjugated goat anti-rabbit IgG antibody (Sigma-Aldrich) was used as the secondary antibody, and signals were detected using Western BLoT Quant HRP Substrate (TaKaRa).

**Statistics and reproducibility**. Statistical analysis was performed using the software Prism 8. Each measurement of enzyme activity was performed thrice (Table 1, Figs. 2f–j and 3e–g, and Supplementary Fig. 4). Error bars indicate the SD of three replicates.

**Reporting summary**. Further information on research design is available in the Nature Research Reporting Summary linked to this article.

## Data availability

Sequence data of FlDet and FlDeg are available from the DDBJ/EMBL/GenBank under the accession number LC519890 and LC519891, respectively. The data for kinetic analysis in Table 1 are shown as Supplementary Data 1. After analyzing raw data for the MS and MS/MS spectra of Fig. 1e on a common computer connected to the mass spectrometer, only the analyzed data were saved in a personal storage. Then, the raw data were overwritten. Therefore, raw data for the MS spectra of Fig. 1e are not available, but the analyzed MS and MS/MS data in this figure are shown as Supplementary Data 2. The source data for the graphs in the main figures are available as Supplementary Data 3. Uncropped and unedited SDS-PAGE gels in main figures are shown as Supplementary Data 4. Other data of this paper are available from the corresponding author upon reasonable request.

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

## Acknowledgements

This study was supported by JSPS KAKENHI (Grant Numbers, 16H04977 and 19H0303909). We thank Dr. Miho Yamada and Dr. Seiko Oka, Open Facility Division, Global Facility Center, Creative Research Institution, Hokkaido University, for performing ESI-MS using an exactive mass spectrometer and providing insight and expertise that considerably assisted the research.

## Author contributions

R.N. and A.I. designed and conducted this experiment, and wrote this paper. T.O. and A.I. supervised this project. Y.O., Y.K. and T.A. were responsible for the NMR analysis.

## Competing interests

The authors declare no competing interests.
