## [Peer Review File. · Communications Biology]

Reviewers' comments:

Reviewer #1 (Remarks to the Author):

Authors proposed novel bacterial pathway for DEHU metabolism which would give impact on the related fields. However, I felt some concerns as below which should be satisfied.

Major concerns:

1. Properties of Fired without Tag.

Authors showed catalytic properties of Fired without Tag is different from those of Fired with Tag, since the novel reaction was found in the Fired without Tag in this study. Most of the characters of Fired seems to be obtained using the Fired with Tag in ref 27. Authors should characterize again the Fired using the Fired without Tag. Especially, it is very important to determine kinetic parameters (K_m , V_{max}) for NAD^+ , $NADP^+$, $NADPH$, $NADH$ and those for KDG in the presence of NAD^+ , $NADP^+$, $NADH$, or $NADPH$. If these values suggest that this enzyme use $NADP^+$ or $NADPH$, authors should re-consider the story of this article and Fig. 4 focusing on only NAD^+ and $NADH$, omitting $NADP^+$ and $NADPH$. Furthermore, kinetic values may give some insight into the problem if the reaction proceeds physiologically.

2. Do the two reactions function physiologically ?

I wonder if the two reactions function physiologically and request some evidence to satisfy my concern, e.g, qPCR of the 5 genes.

3. Structure of KDGR

Authors described that "it was determined to be a compound similar to 2-keto-3-deoxy-D-glucarate (KDGR) (p. 3, lines 93-94). However, after this description, authors regard this compound as KDGR, not a compound similar to KDGR. Further evidence to determine the structure of KDGR is needed.

4. Discussion

The last part of discussion (lines 278-305) sounds poor for me unfortunately. For example:

4-1: The description "Ethanol cannot be practically,,, (lines 282,283)" seems not correct. Refs 43 and 44 showed that the bioengineered E. coli or yeast could practically produce ethanol from both DEHU and mannitol, but these papers seemed not show the ethanol production from DEHU only.

4-2: lines 289-290, Please show the evidence for the original oxidation pathway of DEHU on strain A1. Is there any publication showing this reaction ? Or do authors have some evidence ?

4-3: lines 290, Please show some evidence or description in the text that E. coli and yeast do not have the enzymes for KDGR to alpha-KG conversion.

4-4: It is difficult to understand why the use of these specialized enzymes are the most important (lines 302-305).

Minor concerns.

5: lines 107 -108: Therefore, we attempted to isolate ...may be better.

6: lines 136,154: Enzyme catalyze the reaction, not KDGR, nor keto acids.

7: line 176, Is formation of RP3 correct? I guess consumption of RP3 is better. Please check.

8: line 253, DEHU production in alginate is correct ?

9: line 267, Is sandwiching correct ? Please check.

Reviewer #2 (Remarks to the Author):

Nishiyama et al identified a new pathway for DEHU catabolism. It is established that DEHU can be catabolized by a reductive pathway through kdg followed by kinase and aldolase to produce pyruvate and GAP. The new pathway is using the same first enzyme but in oxidative direction to produce keto deoxy glucarate which is then with two steps converted to alpha keto glutarate.

The manuscript is very well written. It is clear and understandable. The experiments are clearly described. The figures are well designed. The discussion is clear and concise.

The findings are novel and interesting. I was especially surprised to see that the same enzyme can do the oxidation and reduction of the same substrate.

Reviewer #3 (Remarks to the Author):

In this paper, Nishiyama and colleagues report on a novel biochemical pathway for the conversion of DEHU to alpha-ketoglutarate in the alginate-assimilating marine bacterium *Flavobacterium* sp. strain UMI-01. They show that a parallel oxidative pathway is possible besides the reductive route that was already known, and provide evidence that this pathway might be conserved in other alginate-degrading bacteria. This will be of interest for scientists in the field of marine polysaccharide degradation and has potential applications in biorefinery and algal biomass.

Overall, this paper is an elegant, clear and concise report of a yet unknown mechanism for the full assimilation of alginate. The authors characterize *in vitro* a second biochemical function for the previously known enzyme FIRed, and describe two additional enzymes FIDet and FIDeg that sequentially act downstream to yield alpha-ketoglutarate. An important question that remains unanswered in the present paper is whether this parallel oxidative pathway actually takes place *in vivo* and, if yes, in which conditions. To strengthen the conclusions, this study would benefit from functional evidence. For example, are FIDet and FIDeg expressed during alginate degradation? Is the efficiency of alginate utilization higher when both the reductive and oxidative pathways are present? This is important to assess the relevance of the described pathway for alginate catabolism in the environment, and for future applications that authors describe in the discussion. Furthermore, the claim that NADH/NAD⁺ concentrations and ratio are the most important factors regulating the dual pathway (line 203-204) seems speculative with the present data, since this is only based on *in vitro* biochemical activity of enzymes.

In addition, I have some minor comments:

- line 32: typo in "mannuronic"
- line 54: I think NAD(H) should actually be NAD(P)
- line 96: replace "its activity" by "recFIRed activity"
- line 108: change "catalyzing KDGR as a substrate" to "catalyzing KDGR conversion"
- line 122-129: The abnormal mobility of FIDet on SDS-PAGE is intriguing. Could authors propose a potential explanation? Furthermore, the MALDI-TOF-MS shows a signal around 40,000 besides the one at 27,260. Is this significant and, if yes, could it be the form seen on SDS-PAGE?
- line 136: change to "enzymes that catalyze KDGR conversion are lacking"
- line 145: according to the numbering of carbon atoms, I think C-2 and C-3 should actually be C-4 and C-5
- line 154: change to "FIDet can catalyze the potential conversion of alpha-keto acids"
- line 164: the accession number of the draft genome sequence should be mentioned
- line 208: typo in "*Z. galactanivorans*"
- line 220: add "in" between "DEHU" and "strain"
- line 144 and beyond: to my knowledge, transcription as an operon has not been proven for this alginolytic locus in *Flavobacterium* sp. strain UMI-01. Therefore, I would recommend using the word "locus" or "cluster" instead of "operon" in the manuscript.
- line 248-256: what would be the rationale of co-expressing FIDet and FIDeg only with a G-specific lyase, and not with the M-specific lyase? Both G and M-exolytic lyases are expected to yield DEHU.
- line 271-273: this sentence seems a bit disconnected from the rest of the paragraph. What information is it supposed to tell about the tolerance to fluctuations in osmotic pressure?
- line 408: replace "water" by "aqueous"
- Figure 5: percent identities would be more informative than e-value to predict the presence of homologous enzymes in other bacteria

Response to reviewers

Dear reviewers,

We thank you for your valuable comments. We have taken your comments into account and incorporated the new data into the revised manuscript entitled “A novel metabolic pathway for oxidation of 4-deoxy-L-*erythro*-5-hexoseulose uronic acid (DEHU) from alginate in an alginate-assimilating bacterium”. Each comment is answered point by point.

During the revision of this manuscript, since a paper on a novel DEHU reductase in an alginate-synthesizing organism brown alga was published, the relevant sentence was added (lines 49–51). We also corrected some typo errors (lines 214, 363–364, 543, and 831). In the revised manuscript, all of these points, as well as the corrections and additions made according to the reviewers' comments, are indicated in red.

Reviewers' comments:

Reviewer #1 (Remarks to the Author):

Authors proposed novel bacterial pathway for DEHU metabolism which would give impact on the related fields. However, I felt some concerns as below which should be satisfied.

Thank you for spending time to review our manuscript and the helpful comments to better revise this paper. We revised the manuscript along with your comments.

Major concerns:

1. Properties of Flred without Tag.

Authors showed catalytic properties of Flred without Tag is different from those of Flred with Tag, since the novel reaction was found in the Flred without Tag in this study. Most of the characters of Flred seems to be obtained using the Flred with Tag in ref 27. Authors should characterize again the Flred using the Flred without Tag. Especially, it is very important to determine kinetic parameters (K_m , V_{max}) for NAD^+ , $NADP^+$, $NADPH$, $NADH$ and those for KDG in the presence of NAD^+ , $NADP^+$, $NADH$, or $NADPH$. If these values suggest that this enzyme use $NADP^+$ or $NADPH$, authors should re-consider the story of this article and Fig. 4 focusing on only NAD^+ and $NADH$,

omitting NADP⁺ and NADPH. Furthermore, kinetic values may give some insight into the problem if the reaction proceeds physiologically.

We determined K_m and k_{cat} values of NAD⁺, NADP⁺, NADPH, and NADH using FIRed without a His-tag. In addition, those values of the substrate DEHU for FIRed in the presence of NAD⁺, NADP⁺, NADPH, or NADH were determined. These results are shown in Table 1, and Fig. 5 was modified based on these results. Kinetic analysis of FIRed and the physiological regulation of FIRed activity based on these values were described and proposed, respectively (lines 20–22, 136–148, and 262–264).

Table 1. Kinetic parameters of the reductase and oxidase activities of FIRed.

Reductase activity	K_m (mM)	k_{cat} (sec ⁻¹)	k_{cat}/K_m (sec ⁻¹ · mM ⁻¹)
NADH	0.037 ± 0.002	160.0 ± 1.9	4,307
DEHU _{NADH}	2.6 ± 0.1	205 ± 0.7	78.1
NADPH	0.97 ± 0.1	82.5 ± 5.0	84.6
DEHU _{NADPH}	2.7 ± 0.3	87.0 ± 1.8	32.1
Oxidase activity	K_m	k_{cat} (sec ⁻¹)	k_{cat}/K_m (sec ⁻¹ · mM ⁻¹)
NAD ⁺	0.098 ± 0.02	34.8 ± 1.2	354
DEHU _{NAD⁺}	2.4 ± 0.2	53.4 ± 2.0	22.4
NADP ⁺	1.4 ± 0.2	24.6 ± 2.4	18.1
DEHU _{NADP⁺}	2.0 ± 0.1	24.9 ± 2.6	12.3

Fig. 5. Dual metabolic pathways of DEHU in strain UMI-01. Unsaturated monosaccharides are generated from alginate by alginate lyase(s) and are spontaneously or enzymatically⁵⁰ transformed into DEHU. DEHU is oxidized (*upper*) or reduced (*lower*) by FIRed with NAD(P)⁺ or NAD(P)H. Final products of oxidation and reduction pathways are α -KG and pyruvate, respectively.

2. Do the two reactions function physiologically?

I wonder if the two reactions function physiologically and request some evidence to satisfy my concern, e.g, qPCR of the 5 genes.

We prepared specific antibodies for FIRed, FIKin, FIAld, FIDet, and FIDeg and Western blotting analysis was performed using cell extracts of strain UMI-01 cells grown in the medium containing alginate or glucose as the sole carbon source (Fig. 4). These results clarified that alginate is the upregulator for all above enzymes and suggested that each enzyme for reduction and oxidative metabolism of DEHU can function simultaneously *in vivo* (lines 27–28, 247–254).

Fig. 4. Western blot analysis of FIRed, FIKin, FIAld, FIDet, and FIDeg. Strain UMI-01 was cultured in a medium containing glucose (*lanes 1–3*) or alginate (*lanes 4–6*) as the sole carbon source. Samples were prepared from three independent cultures for each condition. *Lane M*, protein marker. **a**, CBB-stained gel; **b**, **c**, **d**, **e**, and **f**, Western blot analysis using anti-FIRed, -FlKin, -FIAld, -FIDet, and -FIDeg antibodies, respectively. The cyan arrowhead indicates the expected migration position of each enzyme. The full-length gel and blots are presented. Blotted filters and overexposed images are shown in Supplementary Fig. 17.

3. Structure of KDGR

Authors described that “it was determined to be a compound similar to 2-keto-3-deoxy-D-glucarate (KDGR) (p. 3, lines 93-94). However, after this description, authors regard this compound as KDGR, not a compound similar to KDGR. Further evidence to determine the structure of KDGR is needed.

We performed the detection of the carboxy group and ESI-MS/MS analysis of RP1 that was predicted as 2-keto-3-deoxy-D-glucarate (KDGR) (lines 96–125). The former experiment suggested that RP1 has a carboxy group that is not derived from the α -keto acid structure (Supplementary Fig. 4). Furthermore, if RP1 cleaves in the process of ionization in the same manner as DEHU, it was reasonable to assume that the molecular species produced by MS/MS of RP1 are derived from the structure of KDGR (Supplementary Fig. 5). We also attempted the structural analysis of RP1 by NMR, but it was difficult to assign each peak because the obtained spectrum was considered to be derived from multiple forms. This is probably because KDGR forms not only a linear structure but also a cyclic structure via an ester bond or a hemiketal bond under the measurement condition. The reason why the number of carboxyl groups per mol of RP1 estimated in Supplementary Fig. 4 was lower than expected could be due to the same reason.

Supplementary Fig. 4. Detection of carboxyl groups using 4-nitrophenyl bromide.

The structures of the examined compounds (D-glucuronate, pyruvate, α -KG, DEHU, and KDGR) and the predicted structure (KDGR) of RP1 are shown. The carboxyl groups in the α -keto acid structures and the other carboxyl groups are highlighted in red and blue, respectively. Values of Δ absorbance were obtained by subtracting the absorbance of each solution containing the indicated compound from that of the buffer. All assays were repeated thrice, and the data are shown as mean \pm S.D.

Supplementary Fig. 5. ESI-MS and MS/MS spectra of DEHU and RP1.

a. ESI-MS spectrum of DEHU. **b.** MS/MS spectrum of the peak at m/z 175.02 in **a.** **c.** ESI-MS spectrum of RP1. **d.** MS/MS spectrum of the peak at m/z 191.02 in **c.** **e.** Possible fragmented structures of DEHU and KDGR. The red-dotted square indicates the β -keto acid structure. Each calculated molecular mass is shown.

4. Discussion

The last part of discussion (lines 278-305) sounds poor for me unfortunately. For example:

4-1: The description “Ethanol cannot be practically,, (lines 282,283)” seems not correct. Refs 43 and 44 showed that the bioengineered E. coli or yeast could practically produce ethanol from both DEHU and mannitol, but these papers seemed not show the ethanol production from DEHU only.

4-2: lines 289-290, Please show the evidence for the original oxidation pathway of DEHU on strain A1. Is there any publication showing this reaction? Or do authors have some evidence?

4-3: lines 290, Please show some evidence or description in the text that E. coli and yeast do not have the enzymes for KDGR to α -KG conversion.

4-4: It is difficult to understand why the use of these specialized enzymes are the most important (lines 302-305).

Thank you for your meaningful comments. We agree with your opinion on this paragraph and have made a major revision, where scientifically poor descriptions and obscure expressions were deleted. In the revised paragraph, the reduction and oxidation of DEHU in bioengineered DEHU-metabolizing microorganisms was considered based on the novel findings obtained in this study (lines 299–311).

Minor concerns.

5: lines 107 -108: Therefore, we attempted to isolate ...may be better.

According to your suggestion, “we isolated” was changed to “we attempted to isolate” (lines 151–152).

6: lines 136,154: Enzymes catalyze the reaction, not KDGR, nor keto acids.

Thank you for pointing out. In the revised manuscript, “enzymes that catalyze KDGR are lacking” and “FIDet can catalyze the potential α -keto acids” was changed to “enzymes that catalyze KDGR conversion are lacking” and “FIDet can catalyze the potential conversion of α -keto acids”, respectively, as suggested by the reviewer #3 (lines 182 and 199).

7: line 176, Is formation of RP3 correct? I guess consumption of RP3 is better. Please check.

We agree with your suggestion and “formation of RP3” was changed to “consumption of RP3” (line 222).

8: line 253, DEHU production in alginate is correct?

Thank you for pointing out. This is our mistake. This sentence should be “DEHU production in strain UMI-01”. However, this paragraph has been rewritten in the revised manuscript, focusing on the expression of DEHU metabolism-related enzymes based on Western blot results (lines 324–341). Therefore, the corresponding sentence has been deleted.

9: line 267, Is sandwiching correct ? Please check.

In the revised manuscript, this paragraph was revised to avoid ambiguous expressions and the sentence including “sandwiching” was deleted.

Reviewer #2 (Remarks to the Author):

Nishiyama et al identified a new pathway for DEHU catabolism. It is established that DEHU can be catabolized by a reductive pathway through kdg followed by kinase and aldolase to produce pyruvate and GAP. The new pathway is using the same first enzyme but in oxidative direction to produce keto deoxy glucarate which is then with two steps converted to alpha keto glutarate.

The manuscript is very well written. It is clear and understandable. The experiments are clearly described. The figures are well designed. The discussion is clear and concise.

The findings are novel and interesting. I was especially surprised to see that the same enzyme can do the oxidation and reduction of the same substrate.

Thank you very much for taking the time to review our manuscript and for your encouragement.

Reviewer #3 (Remarks to the Author):

In this paper, Nishiyama and colleagues report on a novel biochemical pathway for the conversion of DEHU to alpha-ketoglutarate in the alginate-assimilating marine bacterium *Flavobacterium* sp. strain UMI-01. They show that a parallel oxidative pathway is possible besides the reductive route that was already known, and provide evidence that this pathway might be conserved in other alginate-degrading bacteria. This will be of interest for scientists in the field of marine polysaccharide degradation and has potential applications in biorefinery and algal biomass.

Thank you for reviewing our manuscript and giving meaningful comments on the revision of this paper. Each comment is answered point by point.

Overall, this paper is an elegant, clear and concise report of a yet unknown mechanism for the full assimilation of alginate. The authors characterize *in vitro* a second biochemical function for the previously known enzyme F1Red, and describe two additional enzymes F1Det and F1Deg that sequentially act downstream to yield alpha-ketoglutarate. An important question that remains unanswered in the present paper is whether this parallel oxidative pathway actually takes place *in vivo* and, if yes, in which conditions. To strengthen the conclusions, this study would benefit from

functional evidence. For example, are FIDet and FIDeg expressed during alginate degradation? Is the efficiency of alginate utilization higher when both the reductive and oxidative pathways are present? This is important to assess the relevance of the described pathway for alginate catabolism in the environment, and for future applications that authors describe in the discussion. Furthermore, the claim that

NADH/NAD⁺ concentrations and ratio are the most important factors regulating the dual pathway (line 203-204) seems speculative with the present data, since this is only based on in vitro biochemical activity of enzymes.

To answer these comments and their similar comments “1” and “2” of reviewer #1, we performed a kinetic analysis of FIRed (Table 1, and lines 136–148) and investigated protein expression analysis of FIRed, FIKin, FIAld, FIDet, and FIDeg using specific antibodies (Fig. 4, lines 247–254). FIRed could use not only NADH and NAD⁺ but also NADPH and NADP⁺ as coenzymes, but it was found to have a high affinity for the formers. Furthermore, NADH was found to have about 2.6-fold higher affinity than NAD⁺. On the other hand, the affinity for DEHU did not differ much regardless of the type or form of coenzyme. In addition, as a result of Western blotting, it was found that the expression of all five enzymes involved in the reduction and oxidation of DEHU was enhanced by addition of alginate. These results suggested that both reduction and oxidation of DEHU function in the UMI-01 strain and that its regulation depends on the concentration and ratio of NADH/NAD⁺ (lines 262–264).

In addition, I have some minor comments:

- line 32: typo in "mannuronic"

In the revised manuscript, “mannuroninc” was changed to “mannuronic” (line 34).

- line 54: I think NAD(H) should actually be NAD(P)

In the revised manuscript, “NAD(H)” was changed to “NADH” (line 59).

- line 96: replace "its activity" by "recFIRed activity"

In the revised manuscript, “its activity” was replaced to “recFIRed activity” as suggested (line 127).

- line 108: change "catalyzing KDGR as a substrate" to "catalyzing KDGR conversion"

In the revised manuscript, “catalyzing KDGR as a substrate” was changed to “catalyzing KDGR conversion” as indicated (line 152).

- line 122-129: The abnormal mobility of FIDet on SDS-PAGE is intriguing. Could authors propose a potential explanation? Furthermore, the MALDI-TOF-MS shows a signal around 40,000 besides the one at 27,260. Is this significant and, if yes, could it be the form seen on SDS-PAGE?

There are several possible reasons why proteins exhibit abnormal mobility on a SDS-PAGE gel. One of them is that protein has a high isoelectric point (pI). However, since the calculated pI of FIDet is 6.02, it was not considered that the anomalous mobility was due to the pI value. It is also empirically known that the presence of the consecutive basic amino acid sequence can cause abnormal mobility, but no such sequence was found in FIDet. Therefore, we believe that the abnormal mobility of FIDet may be due to the low amount of SDS that can be bound per protein molecule as reported in the report (Arianna et al. (2009) Detergent binding explains anomalous SDS-PAGE migration of membrane proteins. *PNAS*, **106**, 1760–1765), but further experiments are required in the future.

As you pointed out, the MALDI-TOF-MS spectrum showed the peaks of 27,260 and 40,000. Multiple minor peaks may be detected even with samples that appear as a single band in electrophoresis. This is generally thought to be due to the contaminating proteins detected by MALDI-TOF-MS, which is a more sensitive detection method than CBB staining. Therefore, although the possibility that the peak corresponding to 40,000 is derived from FIDet cannot be completely ruled out, we attributed it to another contaminating protein.

- line 136: change to "enzymes that catalyze KDGR conversion are lacking"

The sentence "enzymes that catalyze KDGR are lacking" was changed to "enzymes that catalyze KDGR conversion are lacking" (line 182).

- line 145: according to the numbering of carbon atoms, I think C-2 and C-3 should actually be C-4 and C-5

We agreed with your suggestion. In the revised manuscript, "C-2" and "C-3" were fixed to "C-4" and "C-5", respectively (line 190).

- line 154: change to "FIDet can catalyze the potential conversion of alpha-keto acids"

The sentence "FIDet can catalyze the potential α -keto acids" was changed to "FIDet can catalyze the potential conversion of α -keto acids" (line 199).

-line 164: the accession number of the draft genome sequence should be mentioned

According to your suggestion, the accession number (DDBJ/EMBL/GenBank accession numbers BPLU01000001–BPLU01000077) was mentioned (line 210).

- line 208: typo in "Z. galactanivorans"

We corrected "*Z. galactanovorans*" to "*Z. galactanivorans*" (line 268).

- line 220: add "in" between "DEHU" and "strain"

We have made as you suggested: the word "in" was inserted between "DEHU" and "strain" (line 281).

- line 144 and beyond: to my knowledge, transcription as an operon has not been proven for this alginolytic locus in *Flavobacterium* sp. strain UMI-01. Therefore, I would recommend using the word "locus" or "cluster" instead of "operon" in the manuscript.

Thank you for your suggestion. In the revised manuscript, all "operon" were changed to "cluster" (lines 325, 327, 341, and 792).

- line 248-256: what would be the rationale of co-expressing FIDet and FIDeg only with a G-specific lyase, and not with the M-specific lyase? Both G and M-exolytic lyases are expected to yield DEHU.

As you pointed out, the joint decomposition of the M-exolytic lyase FIAlyB and the G-exolytic lyase FIAlex is considered necessary for the efficient production of DEHU. The gene for FIAlyB is one of members in the conventional alginolytic cluster including genes for the endo-type alginate lyase FIAlyA and a series of enzymes for the DEHU-reduction pathway and FIAlyB is thought to be expressed with FIAlyA, FIRed, FIKin, and FIAld. On the other hand, FIAlex is located away from this conventional cluster and is adjacent to the DEHU oxidative metabolism-related enzymes, FIDet and FIDeg. Western blotting analysis of five DEHU metabolism-related enzymes revealed that FIDet, unlike the other four enzymes, could not be expressed in the absence of alginate. Based on these results, the revised manuscript was focused on for FIRed, FIKin, FIAld, FIDet, and FIDeg expression (lines 324–341) and the description of the expression of exo-type lyases were deleted due to the lack of experimental evidence compared to the DEHU-metabolic enzymes. Although further experiments are required in the future, alginate oligosaccharides or DEHU may activate the transcription of genes belonging to the conventional alginate clusters. Then the resulting DEHU metabolites, such as KDG and KDPG, may promote the transcription of genes for FIDet, FIDeg, and FIAlex. Such a dual regulatory system may contribute to the efficient degradation of alginate to DEHU and the activation of both DEHU-reduction and -oxidation metabolic pathways.

- line 271-273: this sentence seems a bit disconnected from the rest of the paragraph. What information is it supposed to tell about the tolerance to fluctuations in osmotic pressure?

We agreed with your opinion. The second half of this paragraph has been revised as follows (lines 351–357):

"Till date, osmolytes in bacteria, such as amino acids (glutamine, glutamate, and proline), amino acid derivatives (betaine and ectoine), polyhydric alcohols (glycerol and mannitol), and

oligosaccharides (trehalose) have been investigated^{45,46}. We observed that strain UMI-01 was able to grow even in artificial seawater (without Ca²⁺ to avoid alginate gelation) in which alginate was added as the sole carbon source (data not shown). Thus, this bacterium is resistant to high salt concentrations, and glutamate and/or glutamine derived from α -KG may be used as osmolytes in such conditions.”

- line 408: replace "water" by "aqueous"

We have made the replacement as you suggested “water” was replaced to “aqueous” (line 455).

- Figure 5: percent identities would be more informative than e-value to predict the presence of homologous enzymes in other bacteria

This figure was redrawn with the sequence identities as an index according to the proposal and its numbering was changed to Fig. 6. The sentence on lines 274–276 was revised. In addition, two species (*Azotobacter vinelandii* and *Pseudomonas aeruginosa*) were removed because these are not alginate-assimilating bacteria.

Fig. 6. Enzymes involved in DEHU metabolism of strain UMI-01 and their homologous enzymes in other alginate-assimilating bacteria. A BLASTP search was performed using the amino acid sequence of each enzyme of strain UMI-01 as the query.

REVIEWERS' COMMENTS:

Reviewer #1 (Remarks to the Author):

I realize that the revised MS has been greatly improved. Please check the following minor concerns before acceptance.

- 1: Have the revised MS been checked by English-editing service ? If not, the MS should be checked before acceptance.
- 2: line 118: Is "which the same molecular..." OK ?
- 3: Supplementary Fig. 4: I guess RP1, NOT UP1, is correct.
- 4: Authors should discuss if the other DEH reductase catalyzes the oxidation or not. In the ref 31, authors showed that the reductase in brown macroalgae does not show the oxidation in the presence of NAD⁺ or NADP⁺.

Reviewer #2 (Remarks to the Author):

The reviewers comments were all addressed.
It is an excellent manuscript. Congratulations for the good work.

Reviewer #3 (Remarks to the Author):

This manuscript is a revised version of a previous manuscript I reviewed. I thank the authors for taking into account my comments. I feel they have been satisfactorily answered.

Dear reviewers,

We thank all reviewers for taking the time to review our work and their insightful comments that improved the quality and clarity of the manuscript.

REVIEWERS' COMMENTS:

Reviewer #1 (Remarks to the Author): I realize that the revised MS has been greatly improved. Please check the following minor concerns before acceptance.

1: Have the revised MS been checked by English-editing service ? If not, the MS should be checked before acceptance.

Yes. We used academic English editing services. Original manuscript and the revised one were checked by “Cactus Communications” (<https://www.editage.com/>) and “FORTE” (<https://www.fortescience.com/>), respectively.

2: line 118: Is "which the same molecular..." OK ?

Thank you for the suggestion. In the revised manuscript, “which the same molecular mass” was changed to “that have the same molecular mass” (line 118).

3: Supplementary Fig. 4: I guess RP1, NOT UP1, is correct.

Thank you for pointing this out. This is our mistake and “UP1” was changed to “RP1” in Supplementary Fig. 4.

4: Authors should discuss if the other DEH reductase catalyzes the oxidation or not. In the ref 31, authors showed that the reductase in brown macroalgae does not show the oxidation in the presence of NAD⁺ or NADP⁺.

Thank you for your suggestion. In the revised manuscript, we mentioned that none of the previously identified DEHU reductases from bacteria, abalone, and brown alga have been reported to have DEHU oxidation activity, neither recombinant enzymes nor natural proteins. In addition, we discussed that the results of this study suggest that a wild-type protein without any additional sequences such as His-tag should be used to evaluate DEHU oxidoreductase activity (lines 294–305).

Reviewer #2 (Remarks to the Author): The reviewers comments were all addressed. It is an excellent manuscript. Congratulations for the good work.

We are very happy that the reviewer appreciated our research.

Reviewer #3 (Remarks to the Author): This manuscript is a revised version of a previous manuscript I reviewed. I thank the authors for taking into account my comments. I feel they have been satisfactorily answered.

We thank that your valuable comments have improved the quality of our work.